# Mitigating Partial Observability in Sequential Decision Processes via the Lambda Discrepancy

**Cameron Allen**[*]
UC Berkeley[†]

**Aaron Kirtland**[*]
Brown University

**Ruo Yu Tao**[*]
Brown University

**Sam Lobel**
Brown University

**Daniel Scott**
Georgia Tech

**Nicholas Petrocelli**
Brown University

**Omer Gottesman**
Amazon[‡]

**Ronald Parr**
Duke University

**Michael L. Littman**
Brown University

**George Konidaris**
Brown University

## Abstract

Reinforcement learning algorithms typically rely on the assumption that the environment dynamics and value function can be expressed in terms of a Markovian state representation. However, when state information is only partially observable, how can an agent learn such a state representation, and how can it detect when it has found one? We introduce a metric that can accomplish both objectives, without requiring access to—or knowledge of—an underlying, unobservable state space. Our metric, the $\lambda$-discrepancy, is the difference between two distinct temporal difference (TD) value estimates, each computed using TD($\lambda$) with a different value of $\lambda$. Since TD($\lambda$=0) makes an implicit Markov assumption and TD($\lambda$=1) does not, a discrepancy between these estimates is a potential indicator of a non-Markovian state representation. Indeed, we prove that the $\lambda$-discrepancy is exactly zero for all Markov decision processes and almost always non-zero for a broad class of partially observable environments. We also demonstrate empirically that, once detected, minimizing the $\lambda$-discrepancy can help with learning a memory function to mitigate the corresponding partial observability. We then train a reinforcement learning agent that simultaneously constructs two recurrent value networks with different $\lambda$ parameters and minimizes the difference between them as an auxiliary loss. The approach scales to challenging partially observable domains, where the resulting agent frequently performs significantly better (and never performs worse) than a baseline recurrent agent with only a single value network.

## 1 Introduction

The dominant modeling frameworks for reinforcement learning [Sutton and Barto, 2018] define environments in terms of an underlying Markovian state representation. This modeling choice, called the *Markov assumption*, is nearly ubiquitous in reinforcement learning, because it allows environment dynamics, rewards, value functions, and policies all to be expressed as functions that are independent of the past given the most recent state. In principle, an environment can be modeled as either a Markov decision process (MDP) [Puterman, 1994], or its *partially observable* counterpart, a POMDP

---

[*]These authors contributed equally and are ordered alphabetically. Please send any correspondence to camallen@berkeley.edu, aaron_kirtland@brown.edu, and ruoyutao@brown.edu.

[†]Some of this work was completed while at Brown University.

[‡]Work completed while outside of Amazon.

38th Conference on Neural Information Processing Systems (NeurIPS 2024).

[Kaelbling et al., 1998], as long as the underlying state representation contains enough information to satisfy the Markov assumption. The POMDP framework is more general, but POMDPs are typically much harder to solve than MDPs [Zhang et al., 2012], so it is important to know when it is appropriate to use the simpler MDP framework.

Ideally, a system designer can ensure that their reinforcement learning agent is configured to use the appropriate problem model. If an environment is partially observable, the designer may manually add relevant decision-making features to the agent's state representation, either by concatenating observations together [Mnih et al., 2015] or by using other types of feature engineering [Bellemare et al., 2020, Galataud, 2021, Tao et al., 2023]. Alternatively, the designer can manually specify a set of possible environment states over which the agent can maintain a Markovian belief distribution [Kaelbling et al., 1998]. The challenge is that it is not always obvious when the designer has supplied sufficient information to satisfy the Markov assumption. These approaches require the person deploying the agent to know details about the very problem the agent is supposed to help them solve.

An alternative approach—that we explore in this paper—is to let the agent *learn* a good state representation for the problem it is solving. The conventional deep-learning-era wisdom is that the best problem representations come from training a system to solve a task, rather than from human designers. When faced with a potentially partially observable environment, if we provide the agent with a large enough recurrent neural network (RNN), it can perhaps discover an internal state representation "end-to-end" that maximizes reward via gradient descent [Bakker, 2001, Hausknecht and Stone, 2015, Ni et al., 2022, Dong et al., 2022]. Indeed, end-to-end RNN architectures work well for many problems and are both general-purpose and scalable. However, these techniques only implicitly address the problem of learning a Markovian state representation; in this paper, we show that we can achieve much better learning performance when we explicitly tackle the problem of partial observability.

In our approach, the agent learns a state representation by directly minimizing a metric that assesses whether the environment is fully or partially observable. We call our metric the $\lambda$-*discrepancy*, and define it as the difference between two distinct value functions estimated using temporal difference learning (TD) [Sutton, 1988]. Specifically, the TD($\lambda$) method defines a smooth trade-off between one-step TD (i.e., $\lambda = 0$), which makes an implicit Markov assumption, and Monte Carlo (MC) estimation ($\lambda = 1$), which does not, and intermediate $\lambda$ values interpolate between these extremes. By comparing value estimates for two distinct values of $\lambda$, we can check that the agent's observations support Markovian value prediction, and augment them with memory if we find they are incomplete.

Our main contributions[4] are as follows:

1. We introduce and formally define the $\lambda$-discrepancy and prove that it is exactly zero for MDPs and almost always non-zero for a broad class of POMDPs that we characterize. This analysis tells us that our metric reliably *detects* partial observability.

2. We then consider a tabular, proof-of-concept experiment that adjusts the parameters of a memory function to minimize the $\lambda$-discrepancy via gradient descent. For this experiment, we compute the $\lambda$-discrepancy in closed form. Our results demonstrate that minimizing the $\lambda$-discrepancy is a viable path to *mitigating* partial observability.

3. Finally, we integrate our approach into a deep reinforcement learning agent and evaluate on a set of large and challenging POMDP benchmarks. We find that minimizing the $\lambda$-discrepancy between two learned value functions, as an auxiliary loss alongside traditional reinforcement learning, is often significantly more effective (and never worse) than training the same agent with just a single value function.

Overall, we find that the $\lambda$-discrepancy is a simple yet powerful metric for detecting and mitigating partial observability. The metric is also practical, since it can be computed directly from value functions, which are ubiquitous in reinforcement learning. Furthermore, such value functions need only consider observable parts of the environment, so the $\lambda$-discrepancy remains applicable even without the common assumption that the agent knows the full set of possible POMDP states.

---

[4]Code: https://github.com/brownirl/lambda_discrepancy
Videos: https://lambda-discrepancy.github.io

## 2 Background

We consider two frameworks for modeling sequential decision processes: MDPs and POMDPs. The MDP framework [Puterman, 1994] consists of a state space $\mathcal{S}$, action space $\mathcal{A}$, reward function $R : \mathcal{S} \times \mathcal{A} \to \mathbb{R}$, transition function $T : \mathcal{S} \times \mathcal{A} \to \Delta\mathcal{S}$ mapping to a distribution over states, discount factor $\gamma \in [0, 1]$, and initial state distribution $p_0 \in \Delta\mathcal{S}$. The agent's goal is to find a policy $\pi_\mathcal{S} : \mathcal{S} \to \Delta\mathcal{A}$ that selects actions to maximize *return*, $g_t$, the discounted sum of future rewards starting from timestep $t$: $g_t^{\pi_\mathcal{S}} = \sum_{i=0}^{\infty} \gamma^i r_{t+i}$, where $r_i$ is the reward at timestep $i$. We denote the expectation of these returns as *value functions* $V_{\pi_\mathcal{S}}(s) = \mathbb{E}_{\pi_\mathcal{S}}[g_t \mid s_t = s]$ and $Q_{\pi_\mathcal{S}}(s, a) = \mathbb{E}_{\pi_\mathcal{S}}[g_t \mid s_t = s, a_t = a]$.

The POMDP framework [Kaelbling et al., 1998] additionally includes a set of observations $\Omega$ and an observation function $\Phi : \mathcal{S} \to \Delta\Omega$ that describes the probability $\Phi(\omega|s)$ of seeing observation $\omega$ in latent state $s$. POMDPs are a more general model of the world, since they contain MDPs as a special case: namely, when observations have a one-to-one correspondence with states. Similarly, POMDPs where states correspond to disjoint sets of observations are called *block MDPs* [Du et al., 2019]. However, such examples are rare; in typical POMDPs, a single observation $\omega$ does not contain enough information to fully resolve the state $s$. While agents need not *fully* resolve the underlying state to behave optimally, they must retain at least enough information across timesteps that the optimal policy becomes clear.

We are interested in the learning setting, where the agent has no knowledge of the underlying state $s$ nor even the set of possible states $\mathcal{S}$ (let alone the transition, reward, and observation functions). It receives an observation $\omega_t \in \Omega$ at each timestep and must find a way to maximize expected return. One way to do this is to construct a *state representation*, perhaps using some form of memory, on which it can condition its behavior. A state representation $z \in Z$ is *Markovian* if at any timestep $t$, the representation $z_t$ and action $a_t$ together are a sufficient statistic for predicting the reward $r_t$ and next representation $z_{t+1}$, instead of requiring the agent's whole history:

$$\Pr(z_{t+1}, r_t | z_t, a_t) = \Pr(z_{t+1}, r_t | z_t, a_t, \ldots, z_0, a_0). \tag{1}$$

The definition can be applied to states ($s$), observations ($\omega$), and even memory-augmented observations (defined in Sec. 4), by replacing $z$ with the relevant quantity. States and observations are equivalent in MDPs, and this property is satisfied for both by definition, but in POMDPs it typically only holds for the underlying, unobserved state $s$—not the observations.

Markovian state representations have several desirable implications. First, if the Markov property holds then so does the Bellman equation: $V_{\pi_\mathcal{S}}(s) = \sum_{a \in \mathcal{A}} \pi_\mathcal{S}(a \mid s)\big(R(s, a) + \gamma \sum_{s' \in \mathcal{S}} T(s' \mid s, a)V_{\pi_\mathcal{S}}(s')\big)$. The Bellman equation allows agents to estimate the value of a policy, from experiences, and without knowing $T$ or $R$, via a recurrence relation over one-step returns,

$$V_{\pi_\mathcal{S}}^{(i+1)}(s) = \mathbb{E}_{\pi_\mathcal{S}}\left[r_t + \gamma V_{\pi_\mathcal{S}}^{(i)}(s_{t+1}) \mid s_t = s\right], \tag{2}$$

which converges to the unique fixed point $V_{\pi_\mathcal{S}}$. A second implication is that the transition and reward functions, and consequently the value functions $V_{\pi_\mathcal{S}}$ and $Q_{\pi_\mathcal{S}}$, have fixed-sized inputs and are therefore easy to parameterize, learn, and reuse. Finally, it follows from the Markov property that the optimal policy $\pi_\mathcal{S}^*$ can be expressed deterministically and does not require memory [Puterman, 1994].

We can unroll the Bellman equation over multiple timesteps to obtain a similar estimator that uses $n$-step returns: $V_{\pi_\mathcal{S}}(s) = \mathbb{E}_{\pi_\mathcal{S}}[g_{t:t+n} \mid s_t = s]$, where $g_{t:t+n} := r_t + \gamma r_{t+1} + \gamma^2 r_{t+2} + \cdots + \gamma^n V_{\pi_\mathcal{S}}(s_{t+n})$, with $g_{t:t+n} := g_t$ if the episode terminates before $t + n$ has been reached. The same process works for weighted combinations of such returns, including the exponential average:

$$V_{\pi_\mathcal{S}}^\lambda(s) = \mathbb{E}_{\pi_\mathcal{S}}\left[(1 - \lambda) \sum_{n=1}^{\infty} \lambda^{n-1} g_{t:t+n} \mid s_t = s\right], \tag{3}$$

with $V_{\pi_\mathcal{S}}^{\lambda=1}(s) = \mathbb{E}_{\pi_\mathcal{S}}[g_t \mid s_t = s]$ as a special case. Equation (3) defines the TD($\lambda$) value function as an expectation over the so-called $\lambda$-return [Sutton, 1988]. Low values of $\lambda$ give more weight to shorter returns where the value term $V_{\pi_\mathcal{S}}(s_{t+n})$ is discounted less (and thus has greater significance), whereas larger $\lambda$ values put more weight on longer returns where the value term is more heavily discounted. Given an MDP and a fixed policy, the recurrence relations for all TD($\lambda$) value functions share the same fixed point for any $\lambda \in [0, 1]$; however, if the Markov property does not hold, different $\lambda$ may have different TD($\lambda$) fixed points. In this work, we seek to characterize this phenomenon and leverage it for detecting and mitigating partial observability.

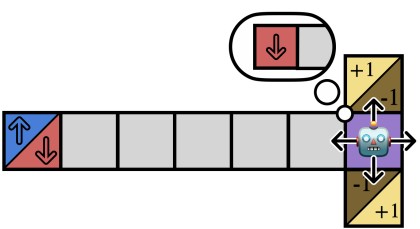

Figure 1: T-maze decision process. The agent must remember the initial observation to earn the maximum reward (+1).

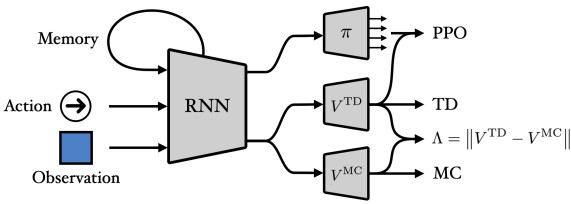

$$\mathcal{L}_{\text{Actor}} = \mathcal{L}_{\text{PPO}} \qquad \mathcal{L}_{\text{Critic}} = \beta \cdot \Lambda + (1 - \beta) \cdot (\mathcal{L}_{V^{\text{TD}}} + \mathcal{L}_{V^{\text{MC}}})$$

Figure 2: $\lambda$-discrepancy augmented network architecture and training objectives.

## 3 Detecting Partial Observability

Before introducing our partial observability metric, let us first consider the T-maze example of Figure 1 under our two candidate decision-making frameworks: MDPs and POMDPs. In the T-maze, the agent only observes the color of its current square, and must remember the start square color (sampled uniformly from **BLUE**/**RED**) to select the action at the junction that leads to the positive reward.

To model the T-maze as a POMDP, the state must include the agent's position and the goal location. The transitions are deterministic; each action moves the agent in the corresponding direction. The goal location is sampled at the start of an episode, and defines both the reward function and the observation for the initial square. The observation function uniquely identifies these starting states, but all corridor states are aliased together (they map to the same observation), as are the two junction states.

We can convert our POMDP model into an MDP model by using observations $\Omega$ as the MDP's "states". This requires new transition and reward functions, $T_\Omega$ and $R_\Omega$, which we define as: $T_\Omega(\omega' \mid \omega, a) := \sum_{s,s' \in \mathcal{S}} \Phi(\omega' \mid s') T_\mathcal{S}(s' \mid s, a) \Pr(s \mid \omega)$ and $R_\Omega(\omega, a) := \sum_{s \in \mathcal{S}} R_\mathcal{S}(s, a) \Pr(s \mid \omega)$, where $\Pr(s \mid \omega)$ is policy-dependent and describes how each hidden state $s_i \in \mathcal{S}$ contributes to the overall environment behavior when we see observation $\omega$. While there are several sensible choices of weighting scheme for this type of state aggregation (uniform over states, relative frequency, etc.), only one of these choices coincides with the distribution $\Pr(s \mid \omega)$. That particular weighting scheme is the one that averages the time-dependent $\Pr(s_t \mid \omega_t)$ over all timesteps, weighted by visitation probability under the agent's policy, and discounted by $\gamma$.[5] We explain how to construct this expression in Appendix C. We call the MDP defined in this way the *effective MDP model* for a given POMDP.

The effective MDP model marginalizes over histories (and POMDP hidden states). For example, the model predicts that going UP from the junction will reach the goal exactly half the time, because the transition dynamics marginalize over (equally likely) starting observations. Note that the POMDP model does no such averaging: if the agent initially observes **BLUE**, then going UP from the junction will *always* reach the goal, but DOWN never will. Thus, we see that, for the T-maze, there is a mismatch between the POMDP model and the effective MDP model. In other words, the POMDP's hidden states $\mathcal{S}$ are a Markovian representation, but its observations $\Omega$ are not, despite the fact that the effective MDP model treats them as Markovian "states".

In principle, an agent could measure partial observability by simultaneously modeling its environment as both a POMDP and an MDP, and comparing the models' predictions. Unfortunately, since the agent lacks any information about the unobserved state space $\mathcal{S}$, the POMDP model would require variable-length history inputs at each time step, and would come with computational and memory costs that grow exponentially with history length. Instead, we propose a model-free approach, using value functions conditioned only on observations, to approximate this comparison in a tractable way.

### 3.1 Value Function Estimation under Partial Observability

The Bellman equation and its sample-based recurrence relations (2) and (3) are defined for Markovian states. If we apply them to the observations of a POMDP, we are actually working with the effective MDP model of that POMDP, instead of the POMDP itself. To see this, consider one-step TD ($\lambda = 0$),

---

[5]One intuitive way to think about $\Pr(s|\omega)$ is that it reflects the relative frequency of trajectories matching both $s$ and $\omega$ versus those just matching $\omega$: $\frac{\mathbb{E}_\tau\left[\sum_{t=0}^\infty \gamma^t \mathbb{1}[s_t=s,\omega_t=\omega]\right]}{\mathbb{E}_\tau\left[\sum_{t=0}^\infty \gamma^t \mathbb{1}[\omega_t=\omega]\right]}$.

where we use the same recurrence relation (2) but now our expectation is sampling from the POMDP:

$$V_\Omega^{\lambda=0}(\omega) = \sum_{s \in \mathcal{S}} \Pr(s \mid \omega) \sum_{a \in \mathcal{A}} \pi(a \mid \omega) \Big( R_S(s, a) + \gamma \sum_{s' \in \mathcal{S}} \sum_{\omega' \in \Omega} \Phi(\omega' \mid s') T_S(s' \mid a, s) V_\Omega^{\lambda=0}(\omega') \Big)$$

$$= \sum_{a \in \mathcal{A}} \pi(a \mid \omega) \Big( R_\Omega(\omega, a) + \gamma \sum_{\omega' \in \Omega} T_\Omega(\omega' \mid a, \omega) V_\Omega^{\lambda=0}(\omega') \Big), \tag{4}$$

where we have suppressed $V_\Omega$'s dependence on the observation-based policy $\pi$ for ease of notation.

We see from Equation (4) that the value function TD computes for a POMDP is the fixed point of the Bellman operator for the effective MDP model.[6] By contrast, the Monte Carlo value estimator ($\lambda = 1$) does not exploit the Bellman equation at all; it simply averages over returns: $V_\pi^{\lambda=1}(s) = \mathbb{E}_\pi[g_t \mid s_t = s]$. Translating the Monte Carlo estimator to POMDPs merely requires one additional expectation to convert from states to observations:

$$V_\Omega^{\lambda=1}(\omega) = \mathbb{E}_\pi[g_t \mid \omega_t = \omega] = \sum_{s \in \mathcal{S}} \Pr(s \mid \omega) \mathbb{E}_\pi[g_t \mid s_t = s] = \sum_{s \in \mathcal{S}} \Pr(s \mid \omega) V_S^{\lambda=1}(s). \tag{5}$$

This means MC ($\lambda = 1$) effectively takes the hidden-state value function $V_S$ of the POMDP and projects it into observation space, whereas TD ($\lambda = 0$) directly computes the value function for the projected model as an MDP, by treating observations as states. Interpolating $\lambda$ between 0 and 1 smoothly varies the value estimate's reliance on the effective MDP model.

For generality, we derive an expression (see Appendix C) for $Q_\pi^\lambda$-values in terms of a given $\lambda$ parameter (expressed in tensor notation for compactness) to reveal how $\lambda$ trades off between the projected state value function and the value function of the projected model:

$$Q_\pi^\lambda = W(I - \gamma T K_\pi^\lambda)^{-1} \vdots R^{\mathcal{SA}}, \text{ where } K_\pi^\lambda = \lambda \Pi^{\mathcal{S}} + (1 - \lambda)\Phi W^\Pi, \tag{6}$$

where the tensor product $\vdots$ contracts two indices instead of one,[7] with tensors defined as follows:

---

$Q_\pi^\lambda$ ($\Omega \times \mathcal{A}$) is a matrix of $Q$-values;
$W$ ($\Omega \times \mathcal{S}$) contains the state-blending weights given by $\Pr(s \mid \omega)$ for observation $\omega$;
$I$ ($\mathcal{S} \times \mathcal{A} \times \mathcal{S} \times \mathcal{A}$) is an identity tensor with $I_{sas'a'} = \mathbb{1}[s = s']\mathbb{1}[a = a']$;
$T$ ($\mathcal{S} \times \mathcal{A} \times \mathcal{S}$) contains the hidden-state transition probabilities $T(s' \mid a, s)$;
$\Pi^{\mathcal{S}}$ ($\mathcal{S} \times \mathcal{S} \times \mathcal{A}$) contains the policy spread over hidden states (see below);
$\Phi$ ($\mathcal{S} \times \Omega$) is the observation function, containing probabilities $\Phi(\omega \mid s)$;
$W^\Pi$ ($\Omega \times \mathcal{S} \times \mathcal{A}$) combines $W$ with $\Pi^{\mathcal{S}}$ to obtain probabilities $\Pr(s, a \mid \omega)$;
$R^{\mathcal{SA}}$ ($\mathcal{S} \times \mathcal{A}$) contains the hidden-state rewards $R(s, a)$.

---

Let us take a moment to parse this equation. First, $\Pi^{\mathcal{S}}$ and $\Phi W^\Pi$ are mappings from states to state-action pairs. We call the former the *MC policy spread*, $\Pi^{\mathcal{S}}_{s,s',a} = \mathbb{1}[s = s'] \sum_\omega \Phi_{s,\omega} \pi_{\omega,a}$, which maps states to the expected policy under their observation distribution. We call the latter the *TD policy spread*, $(\Phi W^\Pi)_{s,s',a} = \sum_\omega \Phi_{s,\omega} \pi_{\omega,a} W_{\omega,s'}$, which reallocates the policy probabilities for a given observation $\omega$ across all states $s'$ that produce that observation, weighted by $W$. $K_\pi^\lambda$ is a convex combination of these two policy spread tensors, parameterized by $\lambda$. Multiplying a policy spread tensor on the left by $T$ produces an ($\mathcal{S} \times \mathcal{A} \times \mathcal{S} \times \mathcal{A}$) transition-policy tensor, describing the probability of each state-action transition $(s, a) \mapsto \Delta(s', a')$ under the policy. Intuitively, Equation (6) says that TD($\lambda$) computes Q-values for a POMDP as though state-action pairs are evolving according to $T K_\pi^\lambda = \lambda T \Pi^{\mathcal{S}} + (1 - \lambda) T \Phi W^\Pi$, which is a mixture of the policy dynamics under two transition models: the MC transition-policy ($T\Pi^{\mathcal{S}}$) and the TD transition-policy ($T\Phi W^\Pi$). The expression $(I - \gamma T K_\pi^\lambda)^{-1} \vdots R$ computes the state-space Q-values for this hybrid transition model. Finally, these are projected through $W$ to compute the observation-space Q-values. (See Appendix C for more details.)

## 3.2 $\lambda$-Discrepancy

We have shown that, under partial observability, $Q_\pi^\lambda$ value functions may differ for different $\lambda$ parameters, due to varying reliance on the effective MDP model in the TD($\lambda$) estimator. We call this difference the $\lambda$-*discrepancy*, and we propose to use it as a measure of partial observability.

---

[6]This equivalence justifies our choice of $\Pr(s|\omega)$ when defining $T_\Omega$ and $R_\Omega$ for the effective MDP model, since states appear in precisely this proportion when we generate experiences in the POMDP.

[7]For 3-dimensional tensors $A$ and $B$, $(AB)_{ijlm} = \sum_k A_{ijk} B_{klm}$, and $(A \vdots B)_{il} = \sum_{jk} A_{ijk} B_{jkl}$.

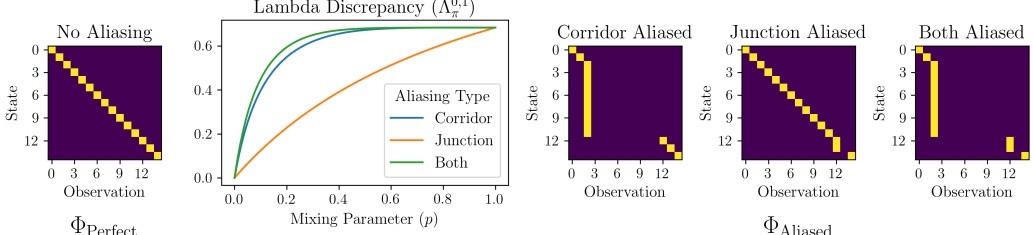

Figure 3: T-Maze $\lambda$-discrepancy, mixing between full and partial observability. (Left) MDP observation function $\Phi_{\mathrm{Perfect}}$. (Right) Various POMDP observation functions $\Phi_{\mathrm{Aliased}}$ that produce aliased observations at the corridor states, junction states, or both. State indices correspond to starting states (0, 1), hallway (2–11), junctions (12, 13), and terminal state (14). Brighter squares indicate higher probability. (Center) $\lambda$-discrepancy has a minimum at zero for full observability and increases with partial observability. We interpolate between perfect observations and aliased ones, where the observation function is $\Phi = (1 - p) \cdot \Phi_{\mathrm{Perfect}} + p \cdot \Phi_{\mathrm{Aliased}}$.

**Definition 1** *For a given POMDP model $\mathcal{P}$ and policy $\pi$, the $\lambda$-discrepancy $\Lambda_{\mathcal{P},\pi}^{\lambda_1,\lambda_2}$ is the weighted norm of the difference between the Q-functions estimated by TD($\lambda$) for $\lambda \in \{\lambda_1, \lambda_2\}$:*

$$\Lambda_{\mathcal{P},\pi}^{\lambda_1,\lambda_2} := \left\| Q_\pi^{\lambda_1} - Q_\pi^{\lambda_2} \right\| = \left\| W\Big( \big(I - \gamma T K_\pi^{\lambda_1}\big)^{-1} - \big(I - \gamma T K_\pi^{\lambda_2}\big)^{-1} \Big) \cdot R^{\mathcal{S}\mathcal{A}} \right\|.$$

The choice of norm can be arbitrary, as can the norm weighting scheme, as long as it assigns positive weight to all reachable observation-action pairs. We discuss choices of weighted norm in Appendix G.2. For brevity, we suppress the $\mathcal{P}$ subscript when the POMDP model is clear from context.

A useful property (that we will prove in Theorem 2) is that if the POMDP has Markovian observations, the $\lambda$-discrepancy is exactly zero for all policies. However, for it to be a useful measure of partial observability, we must also show that the $\lambda$-discrepancy is reliably non-zero when observations are non-Markovian. For this, we have the following theorem.

**Theorem 1** *Given a POMDP model $\mathcal{P}$ and distinct $\lambda \neq \lambda'$, if there exists a policy $\pi : \Omega \to \Delta\mathcal{A}$ such that $\Lambda_{\mathcal{P},\pi}^{\lambda,\lambda'} \neq 0$, then $\Lambda_{\mathcal{P},\pi}^{\lambda,\lambda'} \neq 0$ for all policies except at most a set of measure zero.*

*Proof sketch:* We formulate the $\lambda$-discrepancy of a given policy as the norm of an analytic function, then use the fact that analytic functions are zero everywhere or almost nowhere, along with the fact that norms preserve this property. The full proof is given in Appendix D.

Intuitively, Theorem 1 says that the $\lambda$-discrepancy can detect non-Markovian observations. If it is possible to reveal that a POMDP's observation-space value function does not match that of its effective MDP model, then almost all policies will do so. The theorem further suggests that even if a particular policy has zero $\lambda$-discrepancy, small perturbations to that policy will almost surely detect partial observability if it is present.

We also find that increasing amounts of partial observability lead to larger $\lambda$-discrepancy. In Figure 3 we interpolate between Markovian and aliased T-Maze environments, and consider three types of state aliasing. Here we use a fixed policy that goes RIGHT until the junction, then UP with probability $2/3$ and DOWN w.p. $1/3$. To avoid artifacts due to interactions between the discount factor, weighted norm, and changing observation function, we set $\gamma \approx 1$ and use the (policy-weighted) maximum norm (see Appendix G.2).[8] The results confirm that the $\lambda$-discrepancy is a useful indicator of partial observability, provided that it is non-zero for at least one policy.

### 3.3 What conditions cause the $\lambda$-discrepancy to be zero?

We can characterize the cases in which the $\lambda$-discrepancy is zero for *all* policies by inspecting Definition 1. Because norms are positive definite, it suffices to consider the expression inside the norm. The only ways for this expression to equal zero are either when the difference between policy spread tensors $\big(K_\pi^{\lambda_1} - K_\pi^{\lambda_2}\big)$ is zero (which we will show implies Markovian observations), or it

---

[8]Such artifacts are not a concern in normal environments, since the observation function is fixed.

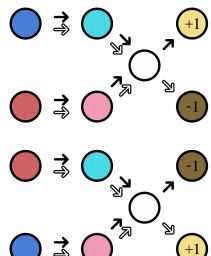 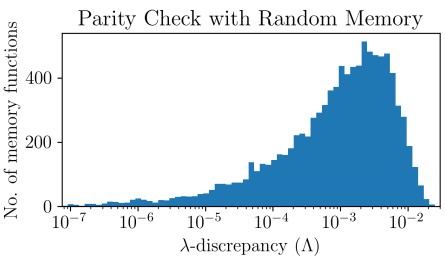 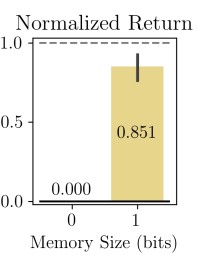

Figure 4: (Left) The Parity Check environment, a POMDP with zero $\lambda$-discrepancy for every policy. (Center) Almost any randomly-initialized 1-bit memory function reveals a $\lambda$-discrepancy. (Right) Memory optimization increases normalized return of subsequent policy learning, whereas memoryless policy optimization fails to beat the uniform random baseline.

is non-zero but is projected away by $T$, $R^{\mathcal{S}\mathcal{A}}$, and/or $W$. We first consider when the policy spread tensors—which are the only terms that depend on $\lambda$—are equal, i.e. $K_\pi^{\lambda_1} = K_\pi^{\lambda_2}$.

**Theorem 2** *For any POMDP $\mathcal{P}$ and any $\lambda_1 \neq \lambda_2$, $K_\pi^{\lambda_1} = K_\pi^{\lambda_2}$ if and only if $\mathcal{P}$ is a block MDP.*

*Proof sketch:* Using Eq. (6), $K_\pi^{\lambda_1} = K_\pi^{\lambda_2}$ can be simplified to $\Pi^{\mathcal{S}} = \Phi W^{\Pi}$, which is satisfied if and only if $\Phi W = I$, i.e. each observation is generated by exactly one hidden state. Proof in Appendix E.

Thus, when the policy spread tensors are equal, it means there is no state aliasing and $\mathcal{P}$ is a block MDP. Now consider the cases where the difference between $K_\pi^{\lambda}$ is projected away by $T$, $R^{\mathcal{S}\mathcal{A}}$, and/or $W$, starting from the policy spread tensors in Definition 1 and working our way outwards.

If the policy spread tensors differ, the transition tensor may project these differences away ($TK_\pi^{\lambda_1} = TK_\pi^{\lambda_2}$), for example, if the transition probabilities leading to the aliased states always occur in some fixed proportion. In such cases, we can collapse the aliased states to non-aliased ones without any loss of information by reformulating the incoming transition probabilities to the aliased states as stochastic transition dynamics (and possibly rewards) *after* the non-aliased states. In other words, $\mathcal{P}$ is equivalent to an MDP. We prove this claim and provide an example of such an environment in Appendix F.

Finally, differences between transition-policies may be projected away by the state-blending weights $W$, the reward function $R^{\mathcal{S}\mathcal{A}}$, or both. $W(I - \gamma TK_\pi^{\lambda_1})^{-1} \colon R^{\mathcal{S}\mathcal{A}} = W(I - \gamma TK_\pi^{\lambda_2})^{-1} \colon R^{\mathcal{S}\mathcal{A}}$. In this case, the effective MDP may be lossy, which is a limitation of the $\lambda$-discrepancy metric. Fortunately, such environments appear to be rare, and they are easy to handle, as we show in the example below.

**Parity Check Environment.** The Parity Check POMDP of Figure 4 has four equally likely starting states, each associated with a pair of colors that the agent will see during the first two timesteps. At the subsequent (white) junction state, the agent must decide based on these colors whether to go UP (black arrow) or DOWN (white arrow). The rewards are defined such that UP is optimal if and only if the color family matches (i.e. RED → PINK or BLUE → CYAN).

The transition-policies for the first eight states differ with and without aliasing, but the differences are perfectly symmetric with respect to rewards and observations.[9] Every observation has zero expected return, and so the $\lambda$-discrepancy is zero for all policies. Our analysis in Appendix H.5 suggests such edge-cases are the exception and not the rule. Even so, such examples are easy to handle: we can simply add a small amount of memory. If the environment is truly an MDP, adding memory should have no effect, but if it is a POMDP, a randomly initialized memory function (defined in the next section) may be enough to break symmetry and produce a $\lambda$-discrepancy (as seen in Figure 4, center).

Taken together, Theorems 1 and 2 suggest that the $\lambda$-discrepancy could be a useful measure for detecting partial observability. In the next section, we demonstrate the efficacy of using the $\lambda$-discrepancy to learn memory functions that mitigate partial observability.

## 4 Memory Learning with the $\lambda$-Discrepancy

The $\lambda$-discrepancy can identify that we need memory, but it cannot yet tell us what to remember. For that, we must replace the POMDP $\mathcal{P}$ in Definition 1 with a memory-augmented version. In general,

---

[9]This environment requires projection by *both* $W$ and $R^{\mathcal{S}\mathcal{A}}$ to zero the differences; neither alone is sufficient.

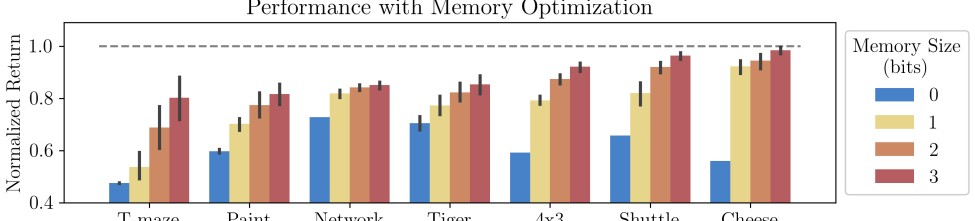

Figure 5: Memory optimization increases normalized return of subsequent policy gradient learning. Performance is calculated as the expected start-state value, and is normalized between a random policy ($y = 0$) and the optimal belief state policy ($y = 1$) found with a POMDP solver [Cassandra, 2003]. Error bars are 95% confidence intervals over 30 seeds.

a memory can be any mapping from agent histories $(\omega_0, a_0, ..., a_{t-1}, r_{t-1}, \omega_t)$ to internal memory states $m$ within some set of memory states $\mathcal{M}$. For practical reasons, we restrict our attention to recurrent memories that update an internal representation incrementally from fixed-size inputs.

We define a *memory function* $\mu : \Omega \times \mathcal{A} \times \mathcal{M} \to \mathcal{M}$ as a mapping from an observation, action, and memory state, $(\omega, a, m)$, to a next memory state $m' = \mu(\omega, a, m)$. Given a POMDP $\mathcal{P}$, a memory function $\mu$ induces a *memory-augmented* POMDP $\mathcal{P}^\mu$, with extended states $\mathcal{S}_\mathcal{M} = \mathcal{S} \times \mathcal{M}$, actions $\mathcal{A}_\mathcal{M} = \mathcal{A} \times \mathcal{M}$, and observations $\Omega_\mathcal{M} = \Omega \times \mathcal{M}$. The augmented transition dynamics $T_\mathcal{M}$ preserve the original transition dynamics $T$ for states $\mathcal{S}$ while allowing the agent full observability and control over memory states $\mathcal{M}$. Memory functions naturally lead to memory-augmented policies $\pi_\mu : \Omega_\mathcal{M} \to \Delta\mathcal{A}_\mathcal{M}$ and value functions $V_{\pi,\mu} : \Omega_\mathcal{M} \to \mathbb{R}$ and $Q_{\pi,\mu} : \Omega_\mathcal{M} \times \mathcal{A}_\mathcal{M} \to \mathbb{R}$ that reflect the expected return under such policies. For details, see Appendix G.1.

The $\lambda$-discrepancy (Definition 1) applies equally well to memory-augmented POMDPs, and can thus be used to determine whether a particular memory function $\mu$ induces a POMDP $\mathcal{P}^\mu$ with a Markovian observation space $\Omega_\mathcal{M}$. We can also use it as a training objective for *learning* such a memory function. We conduct a proof-of-concept experiment on several classic POMDPs for which we can obtain closed-form gradients of the $\lambda$-discrepancy with respect to a parametrized memory function. In each domain, we randomly generate a set of stochastic policies, select the one with maximal $\lambda$-discrepancy $\Lambda_\mathcal{P}^{0,1}$, and adjust the parameters of a memory function $\mu$ to minimize $\Lambda_{\mathcal{P}^\mu}^{0,1}$ via gradient descent. Figure 5 shows the improvement in policy gradient performance due to the resulting memory function for increasing memory sizes. The details of this experiment are provided in Appendix H. We also run the same experiment on the Parity Check example, and provide results in Figure 4 (right).

We see that the $\lambda$-discrepancy can help mitigate partial observability, but it is somewhat inconveniently defined in terms of the closed-form value function fixed-points. Fortunately, with the appropriate choice of weighted norm, we can estimate the $\lambda$-discrepancy purely from sampled observation-action pairs (or, in the case of memory, observation-memory-action tuples):

$$\Lambda_{\mathcal{P},\pi}^{\lambda_1, \lambda_2} := \left\| Q_\pi^{\lambda_1} - Q_\pi^{\lambda_2} \right\| = \left( \mathbb{E}_{(\omega,a)\sim\pi} \left[ \left( Q_\pi^{\lambda_1}(\omega, a) - Q_\pi^{\lambda_2}(\omega, a) \right)^2 \right] \right)^{1/2}. \quad (7)$$

The norm is taken over the on-policy joint observation-action distribution $\Pr(\omega)\pi(a|\omega)$, which allows us to estimate the metric using samples generated by the agent's interaction with the environment. We show how to use it as an optimization objective in the following section.

## 5 A Scalable, Online Learning Objective

So far, we have shown that the $\lambda$-discrepancy can detect partial observability in theory and can mitigate it under certain idealized conditions. Now we demonstrate how to integrate our metric into sample-based deep reinforcement learning to solve problems requiring large, complex memory functions.

### 5.1 Combining the $\lambda$-Discrepancy with PPO

To minimize the $\lambda$-discrepancy, we augment a recurrent version of the proximal policy optimization (PPO) algorithm [Schulman et al., 2017] with an auxiliary loss. We use recurrent PPO as our base algorithm due to its strong performance in many POMDPs [Ni et al., 2022], and since the $\lambda$-discrepancy is a natural extension of generalized advantage estimation [Schulman et al., 2016], which

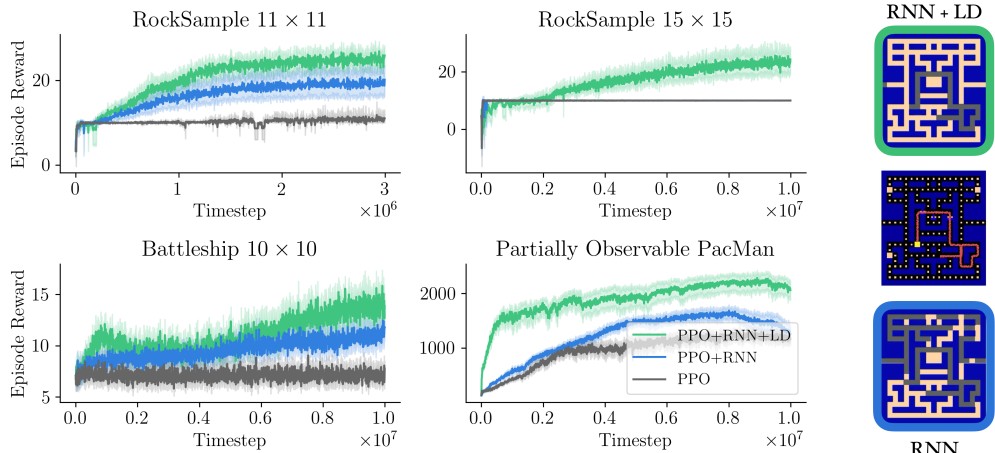

Figure 6: (Left) The $\lambda$-discrepancy auxiliary objective (LD) improves performance over recurrent (RNN) and memoryless PPO. Learning curves shown are the mean and 95% confidence interval over 30 runs. (Right) PacMan memory visualization. The agent moves within the maze (middle), and we reconstruct the dot locations from the agent's memory. RNNs (bottom) benefit from the $\lambda$-discrepancy auxiliary loss (LD, top).

is used in PPO. In this algorithm, a recurrent neural network [Amari, 1972] (specifically a gated recurrent unit, or GRU [Cho et al., 2014]) is used as the memory function $\mu$ that returns a latent state representation given previous latent state and an observation. This latent representation is used as input to an actor network to return a distribution over actions, as well as a critic. The critic is usually a value function network which learns a truncated TD($\lambda$) value estimate for its advantage estimate.

To estimate the $\lambda$-discrepancy, we learn two TD($\lambda$) value function networks with different $\lambda$, parameterized by $\theta_{V,1}$ and $\theta_{V,2}$ respectively, and minimize their mean squared difference as an auxiliary loss:

$$L_\Lambda(\theta) = \mathbb{E}_\pi \left[ \left( V_{\theta_{V,1}}^{\lambda_1}(z_t) - V_{\theta_{V,2}}^{\lambda_2}(z_t) \right)^2 \right], \tag{8}$$

where $z_t = \mu_{\theta_{\mathrm{RNN}}}(\omega_t, z_{t-1})$ is the latent state output by the GRU, and $\theta$ represents all parameters. We train this neural network end-to-end with the standard PPO actor-critic losses and backpropagation through time [Mozer, 1995]. Full details of the algorithm are provided in Appendix I.1.

Any algorithm that uses value functions could, in principle, also leverage the $\lambda$-discrepancy. However, the theoretical results in Section 3 only hold for value function fixed points and not necessarily for estimated values prior to convergence. Inaccurate value functions may lead to an inaccurate $\lambda$-discrepancy. Fortunately, value functions often provide a useful learning signal before convergence, and we observe in the following sections that the $\lambda$-discrepancy is also useful during learning.

## 5.2 Large Partially Observable Environments

We evaluate our approach on a suite of four hard partially observable environments that require complex memory functions. *Battleship* [Silver and Veness, 2010] requires reasoning about unknown ship positions and remembering previous shots. Partially observable *PacMan* [Silver and Veness, 2010] requires localization within the map while tracking dot status and avoiding ghosts with only short-range sensors. *RockSample (11, 11)* and *RockSample (15, 15)* [Smith and Simmons, 2004] have stochastic sensors and require remembering which rocks have been sampled. While these environments were originally used to evaluate partially observable planning algorithms in large-scale POMDPs [Silver and Veness, 2010], we use them to test our sample-based learning algorithms due to the complexity of their required memory functions.[10] See Appendix I.2 for more details.

---

[10]These domains have extremely large state spaces: Battleship has $\sim 2^{100}$ states, without accounting for possible ship locations, and PacMan has $\sim 5^{400}$ states, even when ignoring food dot configurations.

## 5.3 Experiments

We conduct experiments with regular PPO, recurrent PPO, and our $\lambda$-discrepancy-augmented recurrent PPO in Figure 6. We also visualize the agent's memory for the partially observable PacMan environment by reconstructing the dot locations from the RNN latent state to show where the agent "thinks" it has been (see Appendix I.5). We performed a hyperparameter sweep for each method and report learning curves for undiscounted return (see Appendix I.3 for discounted learning curves and Appendix I.4 for additional experimental details).

The $\lambda$-discrepancy objective leads to significantly better final performance and learning rate versus recurrent and memoryless PPO in all environments. In RockSample $(15, 15)$, the baseline agents quickly learn to exit immediately (for +10 reward), but never improve on this. By contrast, minimizing $\lambda$-discrepancy helps the agent learn the missing features that allow it to express better policies. We also run experiments on the classic POMDPs from Section 4, but due to the size of these problems, both baseline and our proposed approach performed almost optimally (see Appendix I.6).

We also observe that the best performing hyperparameters from our sweep offer further evidence for the theory developed in Section 3. Our theory suggests the $\lambda$ parameters should be well separated in order to see the largest $\lambda$-discrepancy. Indeed, we find that a large difference between $\lambda_1$ and $\lambda_2$ was beneficial, with optimal $\lambda$s close to either $0$ or $1$. In addition to this, our hyperparameter sweep also includes a PPO variant with two different TD($\lambda$) value functions but without the $\lambda$-discrepancy auxiliary loss described in Section 5.1. These agents were never selected in the sweep; using the loss in Equation 8 seems to only help with performance in tested environments.

## 6 Related Work

There is an interesting connection between state abstraction [Li et al., 2006], which selectively removes information from state, and partial observability mitigation, which aims to recover state from incomplete observations. Allen et al. [2021] investigated the state abstraction perspective and characterized the properties under which abstract state representations of MDPs either do or do not preserve the Markov property. Several other approaches characterize and measure partial observability and POMDPs. While POMDPs have been shown to be computationally intractable in general [Papadimitriou and Tsitsiklis, 1987], various works have studied complexity measures [Zhang et al., 2012] and defined subclasses with tractable solutions [Littman, 1993, Liu et al., 2022].

The most common strategies for mitigating partial observability are memory-based approaches that summarize history. Early approaches relied on discrete representations of history, including tree representations [McCallum, 1996] or finite-state controllers [Meuleau et al., 1999]. Modern approaches mostly use RNNs [Amari, 1972] trained via backpropagation through time (BPTT) [Mozer, 1995] to tackle non-Markovian decision processes [Schmidhuber, 1990]. Various approaches use recurrent function approximation to learn better state representations. One successful approach is learning a recurrent value function [Lin and Mitchell, 1993, Bakker, 2001, Hausknecht and Stone, 2015] that uses TD error as a learning signal for memory. Policy gradient methods, including PPO (which we compare to), have also been used with recurrent learning to mitigate partial observability [Wierstra et al., 2007, Heess et al., 2015]. Model-based methods can learn a recurrent dynamics model [Hafner et al., 2020] to facilitate planning alongside reinforcement learning. These approaches learn their representations implicitly to improve prediction error, rather than explicitly to mitigate partial observability.

## 7 Conclusion

We introduce the $\lambda$-discrepancy: an observable and differentiable measure of non-Markovianity suitable for mitigating partial observability. The $\lambda$-discrepancy is the difference between two distinct value functions estimated using TD($\lambda$), for two different $\lambda$ values. We characterize the $\lambda$-discrepancy and prove that it reliably distinguishes MDPs from POMDPs. We then use it as a memory learning objective and demonstrate that minimizing it in closed-form helps learn useful memory functions in small-scale POMDPs. Finally, we propose a deep reinforcement learning algorithm which leverages the $\lambda$-discrepancy as an auxiliary loss, and show that it significantly improves the performance of a baseline recurrent PPO agent on a set of large and challenging partially observable tasks.

## Author Contributions

CA, AK and RYT led the project. CA, OG, GK, MLL and SL came up with the initial idea. CA, OG, MLL, SL, and DS conducted the first conceptual investigations and proof-of-concept experiments. AK led the theoretical work, with support from CA, SL, RP, and RYT. RYT led the algorithm development, with support from CA, SL, and GK. CA and RYT led the implementation and experiments, with support from SL, NP, and DS. RP discovered the class of parity check examples and showed that they have zero $\lambda$-discrepancy. CA led the writing, with support from AK and RYT. OG, GK, MLL, SL, and RP advised on the project and provided regular feedback.

## Acknowledgments and Disclosure of Funding

Many thanks to Saket Tiwari, Anita de Mello Koch, Sam Musker, Brad Knox, Michael Dennis, Stuart Russell, and our colleagues at Brown University and UC Berkeley for their valuable advice and discussions towards completing this work. Additional thanks to our reviewers for comments on earlier drafts.

This work was generously supported under NSF grant 1955361 and CAREER grant 1844960 to George Konidaris, NSF fellowships to Aaron Kirtland and Sam Lobel, ONR grant N00014-22-1-2592, a gift from Open Philanthropy to the Center for Human-Compatible AI at Berkeley, and an AI2050 Senior Fellowship for Stuart Russell from the Schmidt Fund for Strategic Innovation.

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

# A  Limitations

The main theoretical limitations of our work come from the class of POMDPs that the $\lambda$-discrepancy is relevant to. The $\lambda$-discrepancy is capable of detecting non-Markovian observations in a "large" class of POMDPs, which we have described using an analytic almost-all characterization. However, there do exist POMDPs in which the fact that observations are non-Markovian cannot be detected by the $\lambda$-discrepancy, and we provide one such example in Section 3.2. The examples of such POMDPs we have found seem to exist on a "knife's edge", meaning that slightly perturbing the parameters of the POMDP results in a non-zero $\lambda$-discrepancy. Furthermore, incorporating memory into the value function seems to reliably produce a $\lambda$-discrepancy, without having to modify the underlying decision process.

The recurrent deep reinforcement learning algorithm we develop is also limited in that it only approximates the $\lambda$-discrepancy. The auxiliary loss in Section 5 is only an approximation of the $\lambda$-discrepancy. It minimizes the difference in value function estimates as opposed to the difference in fixed points that we consider in Section 3. The loss function is therefore a departure from our theoretical analysis, though the empirical results suggest that the value functions need not have fully converged for the discrepancy to be useful. Future work ought to investigate why this works so well in practice.

# B  Broader Impact

As a work studying the foundations of partially observable reinforcement learning, this paper does not have any particular negative societal applications. Application domains could include robotics and tasks such as navigation. The risk for misuse of this work are no different than any other works aimed towards algorithmic improvements for reinforcement leanring. Negative impact mitigation strategies are likewise the same as other similar works.

## C  TD($\lambda$) Fixed Point

Here we derive the fixed point of the TD($\lambda$) action-value update rule in a POMDP, following the Markov version by Sutton [1988]. First, define the expected return given initial observation $\omega_0$ and initial action $a_0$ as

$$
\begin{aligned}
\mathbb{E}_\pi(&G^n|\omega_0, a_0) \\
&= \sum_{s_0} \Pr(s_0|\omega_0) \sum_{r_0} \Pr(r_0|s_0, a_0) r_0 \\
&\quad + \gamma \sum_{s_0} \Pr(s_0|\omega_0) \sum_{s_1} \Pr(s_1|s_0, a_0) \sum_{\omega_1} \sum_{a_1} \Pr(\omega_1|s_1)\Pr(a_1|\omega_1) \sum_{r_1} \Pr(r_1|s_1, a_1) r_1 \\
&\quad + \gamma^2 \sum_{s_0} \Pr(s_0|\omega_0) \sum_{s_1} \Pr(s_1|s_0, a_0) \sum_{\omega_1} \sum_{a_1} \sum_{s_2} \Pr(\omega_1|s_1)\Pr(a_1|\omega_1)\Pr(s_2|s_1, a_1) \\
&\qquad\qquad * \sum_{\omega_2} \sum_{a_2} \Pr(\omega_2|s_2)\Pr(a_2|\omega_2) \sum_{r_2} \Pr(r_2|s_2, a_2) r_2 \\
&\quad + \dots
\end{aligned}
$$

We can define the $n$-step bootstrapped update rule from this given a value matrix $Q_\pi$ by replacing part of the term with coefficient $\gamma^n$ with a $Q_\pi$ value, e.g. for the $n = 2$ case, we get

$$
\begin{aligned}
Q_\pi^{(i+1)}(\omega_0, a_0) &= \sum_{s_0} \Pr(s_0|\omega_0) \sum_{r_0} \Pr(r_0|s_0, a_0) r_0 \\
&\quad + \gamma \sum_{s_0} \Pr(s_0|\omega_0) \sum_{s_1} \Pr(s_1|s_0, a_0) \sum_{\omega_1} \sum_{a_1} \Pr(\omega_1|s_1)\Pr(a_1|\omega_1) \\
&\qquad\qquad\qquad\qquad * \sum_{r_1} \Pr(r_1|s_1, a_1) r_1 \\
&\quad + \gamma^2 \sum_{s_0} \Pr(s_0|\omega_0) \sum_{s_1} \Pr(s_1|s_0, a_0) \sum_{\omega_1} \sum_{a_1} \Pr(\omega_1|s_1)\Pr(a_1|\omega_1) \\
&\qquad\qquad * \sum_{s_2} \Pr(s_2|s_1, a_1) \sum_{\omega_2} \sum_{a_2} \Pr(\omega_2|s_2)\Pr(a_2|\omega_2) Q_\pi^{(i)}(\omega_2, a_2).
\end{aligned}
$$

Translating these expressions into matrix notation, we have

$$
\begin{aligned}
\mathbb{E}_\pi(G^n|\omega_0, a_0) &= \sum_{s_0} W_{\omega_0, s_0} R_{s_0, a_0} \\
&\quad + \gamma \sum_{s_0} W_{\omega_0, s_0} \sum_{s_1} T_{s_0, a_0, s_1} \sum_{\omega_1} \sum_{a_1} \Phi_{s_1, \omega_1} \pi_{\omega_1, a_1} R_{s_1, a_1} \\
&\quad + \gamma^2 \sum_{s_0} W_{\omega_0, s_0} \sum_{s_1} T_{s_0, a_0, s_1} \sum_{\omega_1} \sum_{a_1} \sum_{s_2} \Phi_{s_1, \omega_1} \pi_{\omega_1, a_1} T_{s_1, a_1, s_2} \\
&\qquad\qquad * \sum_{\omega_2} \sum_{a_2} \Phi_{s_2, \omega_2} \pi_{\omega_2, a_2} R_{s_2, a_2} \\
&\quad + \dots,
\end{aligned}
$$

where the terms $W$, $T$, $R$, and $\Phi$ are as in Equation 6, and $\pi$ is the $\Omega \rightarrow \Delta\mathcal{A}$ policy written as an $\Omega \times \mathcal{A}$ tensor with entries in $[0, 1]$. In particular, $W_{\omega, s} = \Pr(s|\omega)$, which averages $\Pr(s_t|\omega_t)$ over all timesteps, weighted by visitation probability and discounted by $\gamma$. This is a well-defined stationary quantity, and it can be computed as follows. First solve the system $Cx = b$ to find the discounted state occupancy counts $x = c(s)$, where $b = p_0$ is the initial state distribution over $\mathcal{S}$, and $C = (I - \gamma(T^\pi)^\top)$ accounts for the policy-dependent state-to-state transition dynamics $T^\pi$ defined by $T^\pi_{s, s'} = \sum_\omega \sum_a \Pr(\omega|s)\Pr(a|\omega)\Pr(s'|s, a)$. Then $\Pr(s|\omega) \propto c(s) * \Phi(\omega|s)$, so we can just multiply these terms together and renormalize. For the 2-step bootstrapped update rule above, we have:

$$
\begin{aligned}
Q_\pi^{(i+1)}(\omega_0, a_0) &= \sum_{s_0} W_{\omega_0, s_0} R_{s_0, a_0} \\
&\quad + \gamma \sum_{s_0} W_{\omega_0, s_0} \sum_{s_1} T_{s_0, a_0, s_1} \sum_{\omega_1} \sum_{a_1} \Phi_{s_1, \omega_1} \pi_{\omega_1, a_1} R_{s_1, a_1} \\
&\quad + \gamma^2 \sum_{s_0} W_{\omega_0, s_0} \sum_{s_1} T_{s_0, a_0, s_1} \sum_{\omega_1} \sum_{a_1} \sum_{s_2} \Phi_{s_1, \omega_1} \pi_{\omega_1, a_1} T_{s_1, a_1, s_2} \\
&\qquad\qquad * \sum_{\omega_2} \Phi_{s_2, \omega_2} \sum_{a_2} \Pi_{\omega_2, a_2} Q_\pi^{(i)}(\omega_2, a_2).
\end{aligned}
$$

Let $\Pi$ be an $\Omega \times \Omega \times \mathcal{A}$ representation of the $\Omega \times \mathcal{A}$ policy $\pi$ with $\Pi_{\omega, \omega', a} = \mathbb{1}[\omega = \omega'] \pi_{\omega, a}$. Likewise, let $\Pi^{\mathcal{S}}$ be the effective policy over latent states, an $\mathcal{S} \times \mathcal{S} \times \mathcal{A}$ representation of the matrix $\Phi \pi$, i.e. $\Pi^{\mathcal{S}}_{s,s',a} = \mathbb{1}[s = s'](\Phi \pi)_{s,a}$. Given a matrix of action-values $Q^{(i)}_{\pi}$, the $n$-step update rule is:

$$Q^{(i+1)}_{\pi} = Q_{\pi, n}(Q^{(i)}_{\pi}) := W \left( \sum_{k=0}^{n-1} (\gamma T \Pi^{\mathcal{S}})^k : R^{\mathcal{S}\mathcal{A}} + \gamma (\gamma T \Pi^{\mathcal{S}})^{n-1} : T \Phi \Pi : Q^{(i)}_{\pi} \right),$$

where $R^{SA}$ is the matrix of reward values as described in Equation 6, and we use $:$ to denote double tensor contraction, while all other tensor products contract a single index.

We also have the standard definition of the TD($\lambda$) update rule as:

$$Q^{(i+1)}_{\pi} = (1 - \lambda) \sum_{n=1}^{\infty} \lambda^{n-1} Q_n(Q^{(i)}_{\pi}).$$

We are concerned with the fixed point of this update rule, which we refer to as $Q^{\lambda}_{\pi}$:

$$Q^{\lambda}_{\pi} = (1 - \lambda) \sum_{n=1}^{\infty} \lambda^{n-1} W \left( \sum_{k=0}^{n-1} (\gamma T \Pi^{\mathcal{S}})^k : R^{\mathcal{S}\mathcal{A}} + \gamma (\gamma T \Pi^{\mathcal{S}})^{n-1} : T \Phi \Pi : Q^{\lambda}_{\pi} \right).$$

Separating this into a reward part with factor $R^{\mathcal{S}\mathcal{A}}$ and a value part with factor $Q^{\lambda}_{\pi}$, we find that the value part is

$$(1 - \lambda) W \left( \sum_{n=1}^{\infty} (\lambda \gamma T \Pi^{\mathcal{S}})^{n-1} \right) : \gamma T \Phi \Pi : Q^{\lambda}_{\pi}$$
$$= (1 - \lambda) W (I - \lambda \gamma T \Pi^{\mathcal{S}})^{-1} : \gamma T \Phi \Pi : Q^{\lambda}_{\pi},$$

and for the reward part, we have the coefficients of $R^{\mathcal{S}\mathcal{A}}$ in the table below for values of $n$ and $k$

|  | $k = 0$ | 1 | 2 | $\ldots$ |
|---|---|---|---|---|
| $n = 1$ | 1 | | | $\ldots$ |
| 2 | $\lambda$ | $\lambda \gamma T \Pi^{\mathcal{S}}$ | | $\ldots$ |
| 3 | $\lambda^2$ | $\lambda^2 \gamma T \Pi^{\mathcal{S}}$ | $\lambda^2 (\gamma T \Pi^{\mathcal{S}})^2$ | $\ldots$ |
| $\vdots$ | $\vdots$ | $\vdots$ | $\vdots$ | $\ddots$ |

where each term is multiplied by $(1 - \lambda) W$ in front. We can then see by summing over rows before columns that the reward part is:

$$(1 - \lambda) W \sum_{k=0}^{\infty} \frac{1}{1 - \lambda} (\lambda \gamma T \Pi^{\mathcal{S}})^k : R^{\mathcal{S}\mathcal{A}}$$
$$= W (I - \lambda \gamma T \Pi^{\mathcal{S}})^{-1} : R^{\mathcal{S}\mathcal{A}}.$$

So we rewrite $Q^{\lambda}_{\pi}$ as follows:

$$Q^{\lambda}_{\pi} = W \left( \left( I^{\mathcal{S}\mathcal{A}} - \lambda \gamma T \Pi^{\mathcal{S}} \right)^{-1} : \left( R^{\mathcal{S}\mathcal{A}} + (1 - \lambda) \gamma T \Phi \Pi : Q^{\lambda}_{\pi} \right) \right).$$

Now let $W^{\mathcal{A}} = W \otimes I^{\mathcal{A}}$, which is $\Omega \times \mathcal{A} \times \mathcal{S} \times \mathcal{A}$, and $W^{\Pi} = \Pi : W^{\mathcal{A}}$, which is $\Omega \times \mathcal{S} \times \mathcal{A}$. Here, $\otimes$ means the Kronecker product. This essentially repeats the $W$ matrix $|\mathcal{A}|$ times to incorporate actions into the tensor. Note that for any $\mathcal{S} \times \mathcal{A}$ tensor $G$, $(W^{\mathcal{A}} : G) = WG$. This is because $(W^{\mathcal{A}} : G)_{\omega a} = \sum_{s, a'} W^{\mathcal{A}}_{\omega a s a'} G_{s a'}$, and the only nonzero terms in this sum are those such that $a = a'$. For these indices, $W^{\mathcal{A}}_{\omega a s a'} = W_{\omega s}$, so $\sum_{s, a'} W^{\mathcal{A}}_{\omega a s a'} G_{s a'} = \sum_s W_{\omega s} G_{s a} = (WG)_{\omega a}$.

Also, let $F = \left( I^{\mathcal{S}\mathcal{A}} - \lambda \gamma T \Pi^{\mathcal{S}} \right)^{-1}$, which is $\mathcal{S} \times \mathcal{A} \times \mathcal{S} \times \mathcal{A}$. Then we find:

$$Q^{\lambda}_{\pi} = W^{\mathcal{A}} : \left( \left( I - \lambda \gamma T \Pi^{\mathcal{S}} \right)^{-1} : \left( R^{\mathcal{S}\mathcal{A}} + (1 - \lambda) \gamma T \Phi \Pi : Q^{\lambda}_{\pi} \right) \right)$$
$$= W^{\mathcal{A}} : \left( F : \left( R^{\mathcal{S}\mathcal{A}} + (1 - \lambda) \gamma T \Phi \Pi : Q^{\lambda}_{\pi} \right) \right)$$
$$= W^{\mathcal{A}} : F : R^{\mathcal{S}\mathcal{A}} + W^{\mathcal{A}} : F : (1 - \lambda) \gamma T \Phi \Pi : Q^{\lambda}_{\pi}.$$

At this point, we can subtract the second term on the right hand side from both sides, factor out $Q_\pi^\lambda$ on the right, and multiply by $\left(I^{\Omega\mathcal{A}} - (1-\lambda)\gamma W^{\mathcal{A}} {:} F {:} T\Phi\Pi\right)^{-1}$ on the left of both sides to obtain:

$$
\begin{aligned}
Q_\pi^\lambda &= \left(I^{\Omega\mathcal{A}} - (1-\lambda)\gamma W^{\mathcal{A}} {:} F {:} T\Phi\Pi\right)^{-1} W^{\mathcal{A}} {:} F {:} R^{\mathcal{SA}} \\
&= W^{\mathcal{A}} {:} \left(I^{\mathcal{SA}} - (1-\lambda)\gamma F {:} T\Phi\Pi {:} W^{\mathcal{A}}\right)^{-1} {:} F {:} R^{\mathcal{SA}} \\
&= W\left(I^{\mathcal{SA}} - (1-\lambda)\gamma F {:} T\Phi W^{\Pi}\right)^{-1} {:} F {:} R^{\mathcal{SA}} \\
&= W\Big(F + (1-\lambda)\gamma F {:} T\Phi W^{\Pi} {:} F + \ldots \\
&\qquad + (1-\lambda)^k \gamma^k (F {:} T\Phi W^{\Pi})^k {:} F + \ldots \Big) {:} R^{\mathcal{SA}},
\end{aligned}
$$

where the last equality follows from expanding the geometric series. Now we use the identity $(A - B)^{-1} = \sum_{k=0}^{\infty}(A^{-1}B)^k A^{-1}$ to find:

$$
\begin{aligned}
Q_\pi^\lambda &= W\left(F^{-1} - (1-\lambda)\gamma T\Phi W^{\Pi}\right)^{-1} {:} R^{\mathcal{SA}} \\
&= W\left(I - \gamma T\left(\lambda\Pi^{\mathcal{S}} + (1-\lambda)\Phi W^{\Pi}\right)\right)^{-1} {:} R^{\mathcal{SA}}.
\end{aligned}
$$

To recap our previous definitions, $W$ is an $\Omega{\times}\mathcal{S}$ tensor, $I$ is an $\mathcal{S}{\times}\mathcal{A}{\times}\mathcal{S}{\times}\mathcal{A}$ tensor, $T$ is an $\mathcal{S}{\times}\mathcal{A}{\times}\mathcal{S}$ tensor, $\Pi^{\mathcal{S}}$ is an $\mathcal{S}{\times}\mathcal{S}{\times}\mathcal{A}$ tensor, $\Phi$ is an $\mathcal{S}{\times}\Omega$ tensor, $W^{\Pi}$ is an $\Omega{\times}\mathcal{S}{\times}\mathcal{A}$ tensor, and $R^{\mathcal{SA}}$ is an $\mathcal{S}{\times}\mathcal{A}$ tensor.

Lastly, we briefly note that one can get the $V_\pi^\lambda$ values by replacing $W$ in the above equation with $W^{\Pi}$ and using a double contraction:

$$
V_\pi^\lambda = W^{\Pi} {:} \left(I - \gamma T\left(\lambda\Pi^{\mathcal{S}} + (1-\lambda)\Phi W^{\Pi}\right)\right)^{-1} {:} R^{\mathcal{SA}}.
$$

We can confirm that $V_\pi^\lambda = \sum_a \pi_{\omega,a} Q_{\omega,a}$, by rewriting the expression on the right as follows:

$$
\begin{aligned}
\sum_a \pi_{\omega,a} Q_{\omega,a} &= \sum_a \pi_{\omega,a} \sum_{s,a'} W^{\mathcal{A}}_{\omega,a,s,a'} B_{s,a'} \\
&= \sum_a \pi_{\omega,a} \sum_s W^{\mathcal{A}}_{\omega,a,s,a} B_{s,a} \\
&= \sum_a \pi_{\omega,a} \sum_s W_{\omega,s} B_{s,a} \\
&= \sum_{s,a} \underbrace{\pi_{\omega,a} W_{\omega,s}}_{W^{\Pi}_{\omega,s,a}} B_{s,a} \\
&= V_\pi^\lambda,
\end{aligned}
$$

where $B = \left(I - \gamma T\left(\lambda\Pi^{\mathcal{S}} + (1-\lambda)\Phi W^{\Pi}\right)\right)^{-1} {:} R^{\mathcal{SA}}$ is an $\mathcal{S} \times \mathcal{A}$ tensor.

# D   Proof of Theorem 1 (Almost All)

In this section we prove Theorem 1, that there is either a $\lambda$-discrepancy for almost all policies or for no policies. Fix $\lambda$ and $\lambda'$. Recall that we define the $\lambda$-discrepancy as follows:

$$\Lambda_\pi := \left\| Q_\pi^\lambda - Q_\pi^{\lambda'} \right\|_{2,\pi} = \left\| W\left( (I - \gamma T K_\pi^\lambda)^{-1} - (I - \gamma T K_\pi^{\lambda'})^{-1} \right) \colon R^{\mathcal{SA}} \right\|,$$

where $K_\pi^\lambda = \lambda \Pi^{\mathcal{S}} + (1-\lambda)\Phi W^\Pi$. Let $Y$ be the largest open set in the space of stochastic $\Omega \times \mathcal{A}$ matrices, considered as a subset of $\mathbb{R}^{|\Omega|(|\mathcal{A}|-1)}$. Now consider the $\lambda$-discrepancy as a function of the policy $\pi$. In other words, we define

$$\Lambda : Y \to \mathbb{R},$$
$$\pi \mapsto Q_\pi^\lambda - Q_\pi^{\lambda'}.$$

Let $X$ be an open subset of $\mathbb{R}^n$. We say that a function $f : X \to \mathbb{R}$ is real analytic on $X$ if for all $x \in X$, $f$ can be written as a convergent power series in some neighborhood of $x$. For this proof, we will utilize the following facts:

1. the composition of analytic functions is analytic [Krantz and Parks, 2002],

2. the quotient of two analytic functions is analytic where the denominator is nonzero,

3. a real analytic function on a domain $X$ is either identically zero or only zero on a set of measure zero [Mityagin, 2020].

We will also use the fact that for an invertible matrix $A$, each entry $A_{ij}^{-1}$ is analytic in the entries of $A$ where the entries of $A$ yield a nonzero determinant. We can prove this by first writing $A^{-1} = \det(A)^{-1} \mathrm{adj}(A) = \det(A)^{-1}\mathrm{cof}(A)^\top$ where $\mathrm{adj}\, A$ is the adjugate of $A$ and $\mathrm{cof}(A)$ is the cofactor matrix of $A$. Each entry of the cofactor matrix is a cofactor that is polynomial in the entries of $A$, and is therefore analytic in them. Therefore, each entry of $A^{-1}$ is the quotient of two analytic functions and is therefore analytic except where $\det A = 0$.

Next, we will show that $\Lambda$ is an analytic function. Note that the variable terms in the equation are $W$, $W^\Pi$, $R^{\mathcal{SA}}$, $T$, $\Pi^{\mathcal{S}}$, and $\Phi$. Of these, $R^{\mathcal{SA}}$, $T$, and $\Phi$ are constant with respect to $\pi$. $\Pi_{ilj}^{\mathcal{S}} = \sum_k \mathbb{1}[i = l]\Phi_{ik}\pi_{kj}$, so each entry of $\Pi^{\mathcal{S}}$ is analytic on $Y$ in the entries of $\pi$. Likewise, $P_{ij} = \sum_{k,a} \Phi_{ik}\pi_{ka}T_{iaj}$ is analytic on $Y$. Therefore, the state-occupancy counts $c = p_0 + \gamma P^\top p_0 + \gamma^2 (P^\top)^2 p_0 + \cdots = (I - \gamma P^\top)^{-1}p_0$, where $p_0$ contains the initial state probabilities, are the composition of analytic functions and thus analytic on $Y$. $W_{\omega s} = \frac{\Phi_{s\omega}c_s}{\sum_{s'} \Phi_{s'\omega}c_{s'}}$ is analytic on $Y$ for the same reason, and the denominator of $W_{\omega s}$, $\sum_{s'} \Phi_{s'\omega}c_{s'}$, is nonzero for all observations able to be observed with nonzero probability. Finally, $\Lambda$ is then a composition of analytic functions on $Y$ and thus analytic itself.

To handle the norm weighting, we note that $w_\pi$ is analytic in $\pi$ as $w_\pi = (1, \pi(a|\omega))$, and the dot product of $w_\pi$ with $\Lambda$ is also analytic. Now, we use the fact mentioned above that the zero set of a nontrivial analytic function is of measure zero. Therefore, the zero set of $\Lambda \cdot w_\pi$ is either zero for all policies or zero only on a set of measure zero. To finish, we note that because norms are positive definite, $\Lambda_\pi = 0$ if and only if $\Lambda \cdot w_\pi = 0$, so this result extends to the normed $\lambda$-discrepancy as well.

# E Proof of Theorem 2 (Block MDP)

In this section, we prove Theorem 2 concerning when the system is a block MDP. Recall that in Equation 6 we define $K_\pi^\lambda = \lambda \Pi^{\mathcal{S}} + (1-\lambda)\Phi W^\Pi$. Suppose $K_\pi^\lambda = K_\pi^{\lambda'}$. We can rewrite this as $(\lambda - \lambda')\Pi^{\mathcal{S}} - (\lambda - \lambda')\Phi W^\Pi = (\lambda - \lambda')(\Pi^{\mathcal{S}} - \Phi W^\Pi) = 0$. This implies that either $\lambda = \lambda'$ or $\Pi^{\mathcal{S}} = \Phi W^\Pi$.

Recall the definition of $\Pi^{\mathcal{S}}$ and $W^\Pi$ in Section C. Expanding the equation $\Pi^{\mathcal{S}} = \Phi W^\Pi$ using these definitions, we have that

$$\Pi_{s,s',a}^{\mathcal{S}} = \mathbb{1}[s = s'](\Phi \pi)_{s,a} = \mathbb{1}[s = s']\sum_\omega \Pr(\omega|s)\Pr(a|\omega);$$

$$(\Phi W^\Pi)_{s,s',a} = \sum_\omega \Phi_{s,\omega} W_{\omega,s',a}^\Pi = \sum_\omega \Pr(\omega|s)\Pr(a|\omega)\Pr(s'|\omega).$$

Then by setting the rightmost expression in each equation equal and simplifying with the indicator function, we have that for all $i, j, k$,

$$\sum_\omega \Pr(\omega|s_i)\Pr(a_k|\omega)\Pr(s_j|\omega) = \begin{cases} \sum_\omega \Pr(\omega|s_i)\Pr(a_k|\omega) & i = j, \\ 0 & i \neq j. \end{cases}$$

We will first consider the case where $i \neq j$. We have that for all $i \neq j$ and all $k$, $\sum_\omega \Pr(\omega|s_i)\Pr(a_k|\omega)\Pr(s_j|\omega) = 0$. Because each term in the sum is nonnegative, this is equivalent to the statement that for all $i \neq j$, all $k$, and all $\omega$, $\Pr(\omega|s_i)\Pr(a_k|\omega)\Pr(s_j|\omega) = 0$. Now note that for all observations $\omega$, there exists some $k'$ such that $\Pr(a_{k'}|\omega) > 0$. Therefore, we have that for all $i \neq j$ and all $\omega$, there exists a $k'$ such that $\Pr(a_{k'}|\omega) > 0$. This implies that for all $i \neq j$ and all $\omega$, $\Pr(\omega|s_i)\Pr(a_{k'}|\omega)\Pr(s_j|\omega) = 0$ and thus $\Pr(\omega|s_i)\Pr(s_j|\omega) = 0$. This means that if state $s_i$ produces an observation $\omega$, then $\omega$ cannot be produced by any other reachable state $s_j \neq s_i$, where two states are said to be *reachable* if there exists a sequence of actions sampled from the policy that enable the agent to reach state $s_i$ from $s_j$ with nonzero probability. In other words, each observation uniquely identifies the hidden state, and the POMDP is a block MDP with corresponding Markovian observations.

Next, we consider the case where $i = j$. We have that for all $i, k$, $\sum_\omega \Pr(\omega|s_i)\Pr(a_k|\omega)(1 - \Pr(s_i|\omega)) = 0$. Because each term is nonnegative, this is equivalent to $\Pr(\omega|s_i)\Pr(a_k|\omega)(\Pr(s_i|\omega) - 1) = 0$. Because we again have that for all observations there exists an action $a_{k'}$ with nonzero probability, this means we can choose $k = k'$ to find $\Pr(\omega|s_i) = 0$ or $\Pr(s_i|\omega) = 1$ for all $\omega, s_i$. This means that either the state $s_i$ does not produce an observation $\omega$, or the observation $\omega$ uniquely determines which state the agent is in. For all $\omega$ and $i = j$, either $\Pr(\omega|s_i) = 0$ or $\Pr(s_j|\omega) = 1$, so if we restrict our focus to the set $\mathcal{O}(s_i) := \{\omega \in \Omega : \Pr(\omega|s_i) > 0\}$, we see that $\sum_{\omega \in \mathcal{O}(s_i)} \Pr(\omega|s_i)\Pr(s_j|\omega) = \sum_{\omega \in \mathcal{O}(s_i)} \Pr(\omega|s_i) \cdot 1 = 1$. For the remaining $\omega \notin \mathcal{O}(s_i)$, $\Pr(\omega|s_i) = 0$.

To recap, the first case tells us that for all $\omega$ and $i \neq j$, $\sum_{\omega \in \Omega} \Pr(\omega|s_i)\Pr(s_j|\omega) = 0$. The second case tells us that for all $\omega$ and $i = j$, $\sum_{\omega \in \Omega} \Pr(\omega|s_i)\Pr(s_j|\omega) = 1$. Combining both these cases, we see that $\sum_\omega \Pr(\omega|s_i)\Pr(s_j|\omega) = \mathbb{1}[s_i = s_j]$, which we can write more succinctly as: $\Phi W = I$. We call $\Phi W$ the state confusion matrix. For block MDPs, observations cause no confusion about which state the agent is in.

Lastly, by going backwards through the proof, we see that the converse is also true. If the system is a block MDP, then $\Pi^{\mathcal{S}} = \Phi W^\Pi$ and so $K_\pi^\lambda = K_\pi^{\lambda'}$.

## F    Proof that matching transition-policies is equivalent to an MDP.

Assume that $TK = TK'$. We can reduce this as follows:

$$0 = TK - TK' = T(\lambda \Pi^S + (1-\lambda)\Phi W^\Pi) - T(\lambda'\Pi^S + (1-\lambda')\Phi W^\Pi) = (\lambda - \lambda')T(\Pi^S - \Phi W^\Pi)$$

which implies either $\lambda = \lambda'$ or $T\Pi^S = T\Phi W^\Pi$.

Claim: if $\mathcal{P}$ is a POMDP such that $T\Pi^S = T\Phi W^\Pi$ for any policy, then $\mathcal{P}$ is equivalent to an MDP.

Recall that the transition function of the effective MDP model is $WT\Phi$, and the rewards are $WR^{\mathcal{SA}}$. Intuitively, if $T$ and $WT\Phi$ are equivalent transition functions, it means they induce the same visitation after any number of steps. Note that due to the premise, the following quantities are equal:

$$WI^{\mathcal{SA}} = I^{\Omega\mathcal{A}} \colon W^A;$$
$$W(T\Pi^S) = W(T\Phi W^\Pi) = WT\Phi\Pi \colon W^A = (WT\Phi\Pi) \colon W^A;$$
$$W(T\Pi^S)^2 = W(T\Pi^S \colon T\Pi^S) = W(T\Phi W^\Pi \colon T\Phi W^\Pi) = W(T\Phi\Pi \colon W^A \colon T\Phi\Pi \colon W^A)$$
$$= W(T\Phi\Pi \colon WT\Phi\Pi \colon W^A) = (WT\Phi\Pi) \colon (WT\Phi\Pi) \colon W^A = (WT\Phi\Pi)^2 \colon W^A;$$
$$W(T\Pi^S)^3 = W(T\Pi^S \colon T\Pi^S \colon T\Pi^S) = W(T\Phi W^\Pi \colon T\Phi W^\Pi \colon T\Phi W^\Pi)$$
$$= W(T\Phi\Pi \colon W^A \colon T\Phi\Pi \colon W^A \colon T\Phi\Pi \colon W^A) = W(T\Phi\Pi \colon WT\Phi\Pi \colon WT\Phi\Pi \colon W^A)$$
$$= (WT\Phi\Pi) \colon (WT\Phi\Pi) \colon (WT\Phi\Pi) \colon W^A = (WT\Phi\Pi)^3 \colon W^A;$$

and so on.

Multiplying each successive equation by $\gamma^i$ and summing the results gives:

$$W(I^{\mathcal{SA}} - \gamma T\Pi^S)^{-1} = (I^{\Omega\mathcal{A}} - \gamma(WT\Phi)\Pi)^{-1} \colon W^A.$$

For any given observation, the POMDP model and the effective MDP model make the same predictions about the expected discounted visitation to future state-action pairs. In other words, they have the same successor representation.

Multiplying both sides on the right by an arbitrary reward function $R^{SA}$,

$$W(I^{\mathcal{SA}} - \gamma T\Pi^S)^{-1} \colon R^{\mathcal{SA}} = (I^{\Omega\mathcal{A}} - \gamma(WT\Phi)\Pi)^{-1} \colon W^A \colon R^{\mathcal{SA}}$$
$$= (I^{\Omega\mathcal{A}} - \gamma(WT\Phi)\Pi)^{-1} \colon (WR^{\mathcal{SA}}),$$

we see that the MC value according to the POMDP model (left hand side) is equal to the TD value under the effective MDP model (right hand side). Both models make the same value predictions for any reward function, therefore the models are equivalent.

We show an example of such a POMDP and its equivalent MDP in Figure 7.

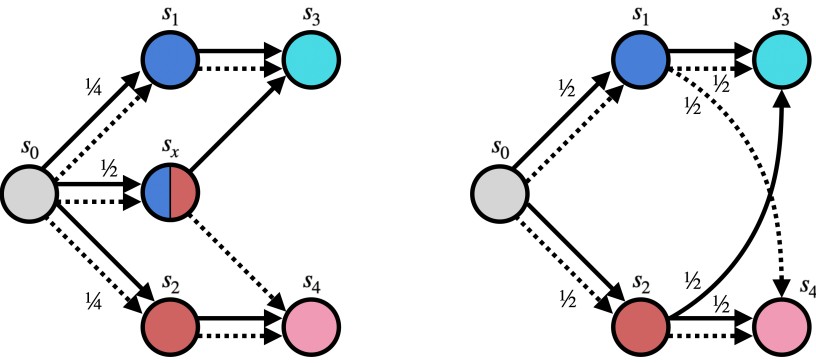

Figure 7: (Left) A POMDP with six states and two actions, where $T\Phi W^\Pi = T\Pi^S$. From state $s_0$, both actions transition to states $s_1$, $s_x$, and $s_2$ with the labeled probabilities. State $s_x$ produces two observations (equivalent to those of $s_1$ and $s_2$) with equal probability. (Right) An equivalent five-state MDP model that behaves identically to the POMDP for any policy.

# G    Memory Augmentation and Norm Details

In this section, we clarify theoretical details in augmenting a POMDP with memory and the $\lambda$-discrepancy norm.

## G.1    Memory-Augmented POMDP

As referenced in Section 4, here we will explain how to define a memory-augmented POMDP from a base POMDP $(\mathcal{S}, \mathcal{A}, T, R, \Omega, \Phi, \gamma)$. Given a set of memory states $\mathcal{M}$, we will augment the POMDP as follows:

$$
\begin{aligned}
\mathcal{S}_{\mathcal{M}} &:= \mathcal{S} \times \mathcal{M} \\
\mathcal{A}_{\mathcal{M}} &:= \mathcal{A} \times \mathcal{M} \\
\Omega_{\mathcal{M}} &:= \Omega \times \mathcal{M} \\
T_{\mathcal{M}} &: \mathcal{S}_{\mathcal{M}} \times \mathcal{A}_{\mathcal{M}} \to \Delta \mathcal{S}_{\mathcal{M}}; & (s_0, m_0, a_1, m_1) &\mapsto T(\cdot | s_0, a_1) \times \delta_{m_1}(\cdot) \\
R^{\mathcal{S}_{\mathcal{M}} \mathcal{A}_{\mathcal{M}}} &: \mathcal{S}_{\mathcal{M}} \times \mathcal{A}_{\mathcal{M}} \to \mathbb{R}; & (s_0, m_0, a_1, m_1) &\mapsto R(s_0, a_1) \\
\Phi_{\mathcal{M}} &: \mathcal{S}_{\mathcal{M}} \to \Delta \Omega_{\mathcal{M}}; & (s_0, m_0) &\mapsto \Phi(\cdot \mid s_0) \times \delta_{m_0}(\cdot) \\
\gamma_{\mathcal{M}} &:= \gamma,
\end{aligned}
$$

where $\delta_{x_0}(\cdot)$ is the discrete point distribution centered at $x_0$, and $\times$ between two distributions denotes the product distribution. To demonstrate what this means, the notation above is equivalent to seeing $T_{\mathcal{M}}$ instead as a function mapping to $[0, 1]$ and defining $T_{\mathcal{M}} : \mathcal{S}_{\mathcal{M}} \times \mathcal{A}_{\mathcal{M}} \times \mathcal{S}_{\mathcal{M}} \to [0, 1], (s_0, m_0, a_1, m_1, s_2, m_2) \mapsto T(s_2 | s_0, a_1) \mathbb{1}[m_1 = m_2]$, where a memory-augmented action $(a_1, m_1)$ means to take action $a_1$ and set the memory state to $m_1$.

This augmentation scheme uses the memory states $\mathcal{M}$ in three ways: as augmentations of states, actions, and observations. The state augmentation concatenates the environment state $\mathcal{S}$ with the agent's internal memory state $\mathcal{M}$. Meanwhile, the action augmentation $\mathcal{A}_{\mathcal{M}}$ provides the agent with a means of managing its internal memory state. Together, these augmentations allow writing the augmented transition dynamics $T_{\mathcal{M}}$, which are defined so as to preserve the underlying state-transition dynamics $T$ while allowing the agent full control to select its desired next memory state. The observation augmentation $\Omega_{\mathcal{M}}$ provides the agent with additional context with which to make policy decisions, and the observation function $\Phi_{\mathcal{M}}$ preserves the original behavior of the observation function $\Phi$ while giving the agent complete information about the internal memory state.

We define an augmented policy $\pi_{\mathcal{M}}$ and its tensor counterparts $\Pi_{\mathcal{M}}$ and $\Pi^{\mathcal{S}_{\mathcal{M}}}$ over observations and states, respectively, as follows:

$$
\begin{aligned}
\pi_{\mathcal{M}} &: \Omega_{\mathcal{M}} \to \Delta \mathcal{A}_{\mathcal{M}}; \\
\Pi_{\mathcal{M}} &\text{ is of size } (\Omega_{\mathcal{M}} \times \Omega_{\mathcal{M}} \times \mathcal{A}_{\mathcal{M}}); \\
\Pi^{\mathcal{S}_{\mathcal{M}}} &\text{ is of size } (\mathcal{S}_{\mathcal{M}} \times \mathcal{S}_{\mathcal{M}} \times \mathcal{A}_{\mathcal{M}});
\end{aligned}
$$

where

$$
\Pi_{\mathcal{M}}(\omega_0, m_0, \omega_1, m_1, a_2, m_2) = \mathbb{1}[(\omega_0, m_0) = (\omega_1, m_1)] \pi_{\mathcal{M}}(a_2, m_2 | \omega_1, m_1);
$$
$$
\Pi^{\mathcal{S}_{\mathcal{M}}}(s_0, m_0, s_1, m_1, a_2, m_2) = \mathbb{1}[(s_0, m_0) = (s_1, m_1)] \sum_{(\omega, m) \in \Omega_{\mathcal{M}}} \Phi_{\mathcal{M}}(\omega, m | s_0, m_0) \pi_{\mathcal{M}}(a_2, m_2 | \omega, m).
$$

We extend the $\Pr(s | \omega)$ tensors to $\mathcal{S}_{\mathcal{M}}$ and $\Omega_{\mathcal{M}}$ as follows:

$$
\begin{aligned}
W_{\mathcal{M}} &\text{ is of size } (\Omega_{\mathcal{M}} \times \mathcal{S}_{\mathcal{M}}); \\
W_{\mathcal{M}}^{\Pi} &\text{ is of size} (\Omega_{\mathcal{M}} \times \mathcal{S}_{\mathcal{M}} \times \mathcal{A}_{\mathcal{M}});
\end{aligned}
$$

where

$$
W_{\mathcal{M}}(\omega_0, m_0, s_1, m_1) = \mathbb{1}[m_0 = m_1] \Pr(s_1 | \omega_0);
$$
$$
W_{\mathcal{M}}^{\Pi}(\omega_0, m_0, s_1, m_1, a_2, m_2) = \mathbb{1}[m_0 = m_1] \Pr(s_1 | \omega_0) \pi_{\mathcal{M}}(a_2, m_2 | \omega_0, m_0).
$$

One appealing aspect of the above augmentation process is that it makes the agent's control over its memory function explicit, via "memory-augmented actions" $\mathcal{A}_\mathcal{M} = \mathcal{A} \times \mathcal{M}$. However, it is perhaps a bit unusual to combine the agent's policy and memory together. An equivalent and perhaps more intuitive formulation decomposes this action-memory policy into a separate action policy $\pi_\mu : \Omega \times \mathcal{M} \to \Delta\mathcal{A}$ and a memory function $\mu : \Omega \times \mathcal{M} \times \mathcal{A} \to \Delta\mathcal{M}$:

$$\pi_\mathcal{M}(a, m' \mid \omega, m) = \pi_\mu(a \mid \omega, m)\mu(m' \mid \omega, m, a).$$

Note that we define memory functions as distributions of next memory states for generality.

There are two ways to view this augmentation procedure. In the first view, outlined above, the agent selects both an action $a$ and a next memory state $m'$, and the environment $T_\mathcal{M}$ responds by sampling the next state $s' \sim T(\cdot|s, a)$ while deterministically setting the next memory state to $m'$. In the second (equivalent) view, the agent selects only action $a$, with the agent's memory function behavior folded into the transition dynamics $T_\mu$. This view leads to the following POMDP quantities:

$$T_\mu : \mathcal{S}_\mathcal{M} \times \mathcal{A} \to \Delta\mathcal{S}_\mathcal{M}; \qquad (s_0, m_0, a_0) \mapsto T(\cdot \mid s_0, a_0)\mu_\mathcal{S}(\cdot \mid s_0, m_0, a_0),$$
$$R_\mu^{\mathcal{S}_\mathcal{M}\mathcal{A}} : \mathcal{S}_\mathcal{M} \times \mathcal{A} \to \mathbb{R}; \qquad (s_0, m_0, a_0) \mapsto R(s_0, a_0),$$

where $\mu_\mathcal{S}(\cdot|s_0, m_0, a_0) := \sum_{\omega \in \Omega} \Phi(\omega|s_0)\mu(\cdot|\omega_0, a_0, m_0)$ is the effective memory function for states.

The $\mu$-specific policy and weight tensors are defined as follows:

$$\Pi_\mu \text{ is of size } (\Omega_\mathcal{M} \times \Omega_\mathcal{M} \times \mathcal{A});$$
$$\Pi_\mu^{\mathcal{S}_\mathcal{M}} \text{ is of size } (\mathcal{S}_\mathcal{M} \times \mathcal{S}_\mathcal{M} \times \mathcal{A});$$
$$W_\mu^\Pi \text{ is of size} (\Omega_\mathcal{M} \times \mathcal{S}_\mathcal{M} \times \mathcal{A});$$

where

$$\Pi_\mu(\omega_0, m_0, \omega_1, m_1, a_2) = \mathbb{1}[(\omega_0, m_0) = (\omega_1, m_1)]\pi_\mu(a_2|\omega_1, m_1);$$
$$\Pi_\mu^{\mathcal{S}_\mathcal{M}}(s_0, m_0, s_1, m_1, a_2) = \mathbb{1}[(s_0, m_0) = (s_1, m_1)] \sum_{(\omega,m) \in \Omega_\mathcal{M}} \Phi_\mathcal{M}(\omega, m|s_0, m_0)\pi_\mu(a_2|\omega, m);$$

$$W_\mu^\Pi(\omega_0, m_0, s_1, m_1, a_2) = \mathbb{1}[m_0 = m_1]\Pr(s_1|\omega_0)\pi_\mu(a_2|\omega_0, m_0).$$

To recap, the first augmentation scheme transforms the POMDP $\mathcal{P} = (\mathcal{S}, \mathcal{A}, \Omega, T, R^{\mathcal{S}\mathcal{A}}, \Phi, p_{\mathcal{M}_0}, \gamma)$ into $\mathcal{P}_\mathcal{M} = (\mathcal{S}_\mathcal{M}, \mathcal{A}_\mathcal{M}, \Omega_\mathcal{M}, T_\mathcal{M}, R^{\mathcal{S}_\mathcal{M}\mathcal{A}_\mathcal{M}}, \Phi_\mathcal{M}, p_{\mathcal{M}_0}, \gamma)$. The second view forms an equivalent POMDP $\mathcal{P}_\mu = (\mathcal{S}_\mathcal{M}, \mathcal{A}, \Omega_\mathcal{M}, T_\mu, R_\mu^{\mathcal{S}_\mathcal{M}\mathcal{A}}, \Phi_\mathcal{M}, \gamma)$, which folds the agent's memory function $\mu$ into the transition dynamics.

Since both of these are valid POMDPs and we already have a general expression for the TD($\lambda$) value function of a POMDP, we can immediately write down the corresponding value functions for each of these types of augmentation. For $\mathcal{P}_\mathcal{M}$, we have:

$$Q_{\pi_\mathcal{M}}^\lambda = W_\mathcal{M}\left(I^{\mathcal{S}_\mathcal{M}\mathcal{A}_\mathcal{M}} - \gamma T_\mathcal{M}\left(\lambda\Pi^{\mathcal{S}_\mathcal{M}} + (1 - \lambda)\Phi_\mathcal{M}W_\mathcal{M}^\Pi\right)\right)^{-1}:R^{\mathcal{S}_\mathcal{M}\mathcal{A}_\mathcal{M}}; \qquad (9)$$

and for $\mathcal{P}_\mu$, we have:

$$Q_{\pi_\mu}^\lambda = W_\mathcal{M}\left(I^{\mathcal{S}_\mathcal{M}\mathcal{A}} - \gamma T_\mu\left(\lambda\Pi_\mu^{\mathcal{S}_\mathcal{M}} + (1 - \lambda)\Phi_\mathcal{M}W_\mu^\Pi\right)\right)^{-1}:R_\mu^{\mathcal{S}_\mathcal{M}\mathcal{A}}. \qquad (10)$$

We provide pseudocode for taking this memory-Cartesian product of a given POMDP in Appendix H.3, Algorithm 3. The pseudocode uses the view that aligns with Equation (10), since that view matches our implementation.

### G.2 $\lambda$-Discrepancy Norm Weighting

The $\lambda$-discrepancy as introduced in Definition 1 contains a weighted norm $\|C \cdot x\|_p$ over the observations $\omega$ and actions $a$ of the decision process. We use several choices of norm $p$ and weighting $C$ in our experiments. For $p$, we use both the $p = 2$ or $L^2$ norm, and the $p = \infty$ or max norm, as described below. For $C$, we have policy-weighted and occupancy weighted cases. In the policy-weighted case,

we assign the $(\omega, a)$ entry of $C$ the weight $(1, \pi(a|\omega))$. In the occupancy-weighted case, we assign the $(\omega, a)$ entry of $C$ the weight $(\Pr(\omega), \pi(a|\omega))$, where $\Pr(\omega)$ is proportional to the discounted observation occupancy.

In Figure 4 and Figure 5, we use the policy-weighted $L^2$ norm. We use this norm due to issues with performing closed-form gradient descent on the T-maze POMDP. In T-maze, states are aliased in two ways: aliasing between the two hallways (goal up or down), and aliasing between the hallway states. Occupancy weighting puts more weight on discriminating between the hallway states, since the hallway observations appear more frequently for most policies. Uniformly weighting the $\lambda$-discrepancy puts more weight on the initial observations, which results in memory functions that place more emphasis on resolving $\lambda$-discrepancy due to the starting observations. Closed-form experiments with occupancy weighting yielded less consistent results for T-maze. This seems to only be an issue for closed-form calculation of $\lambda$-discrepancy, as we show in Appendix I.6 that occupancy weighting in the sample-based setting does not detract from performance in these environments.

In Figure 3, we use the occupancy-weighted max norm to avoid artifacts due the changing observation function (see text in Section 3.2).

In our sample-based settings in Figures 6, 11, 12, and 14, and Table 2, we use the occupancy-weighted $L^2$ norm, since it is the simplest choice when using samples as it does not require re-weighting or importance sampling.

Regardless of which norm we choose, we can estimate gradients as in Equation 7 because:

$$
\begin{aligned}
\Lambda_{\mathcal{P},\pi}^{\lambda_1,\lambda_2} &:= \left\| Q_\pi^{\lambda_1} - Q_\pi^{\lambda_2} \right\| \\
&= \left( \sum_{\omega,a} \Pr(\omega) \Pr(a|\omega)(Q_\pi^{\lambda_1}(\omega,a) - Q_\pi^{\lambda_2}(\omega,a))^2 \right)^{1/2} \\
&= \left( \sum_{\omega,a} \Pr(\omega,a)(Q_\pi^{\lambda_1}(\omega,a) - Q_\pi^{\lambda_2}(\omega,a))^2 \right)^{1/2} \\
&= \left( \mathbb{E}_{(\omega,a)\sim\pi} \left[ \left( Q_\pi^{\lambda_1}(\omega,a) - Q_\pi^{\lambda_2}(\omega,a) \right)^2 \right] \right)^{1/2}.
\end{aligned}
$$

In practice, when optimizing the $\lambda$-discrepancy, we omit the outer square root and simply compute the mean squared difference between Q-values, since the square-root function is a monotonic transformation and its argument is non-negative.

## H Environments and Experimental Details for Closed-Form Memory Optimization

We now describe the proof-of-concept experiments from Section 4 involving closed-form memory optimization on a range of small-scale classic POMDPs. The environments are: T-maze [Bakker, 2001], the Tiger problem [Cassandra et al., 1994], Paint [Kushmerick et al., 1995], Cheese Maze, Network, Shuttle [Chrisman, 1992], and the partially observable version of the $4 \times 3$ maze [Parr and Russell, 1995].

We first begin by detailing the small-scale POMDPs we use. Then we describe the algorithm used to calculate closed-form $\lambda$-discrepancy gradients, as well as procedures for policy improvement after learning a memory function. Apart from the T-maze, all other POMDPs used in Section 4 were taken from pre-defined POMDP definitions [Cassandra, 2003].

We made one slight modification to the Tiger environment that preserves the original environment behavior but adapts the domain specification to match our formalism such that observations are only a function of state. The original Tiger domain used a hand-coded initial belief distribution that was uniform over the two states L/R, and did not emit an observation until after the first action was selected. Thereafter, the observation function was action-dependent, with state-action pair (L, listen) emitting observations left and right with probability $0.85$ and $0.15$ respectively, and other actions (L, *) emitting uniform observations and returning to the initial belief distribution. Since our agent does not have access to the set of states, it cannot use an initial belief distribution. To achieve the same behavior, we modified the domain by splitting each state L/R into an initial state $L_1$/$R_1$ that always emits an initial observation, and a post-listening state $L_2$/$R_2$ that uses the $0.85$/$0.15$ probabilities. We visualize these changes in Figure 8. This type of modification is always possible for finite POMDPs and does not change the underlying dynamics.

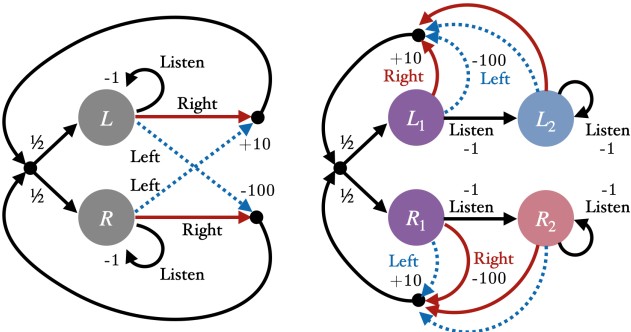

Figure 8: Visualizations of the Tiger POMDP. In the original version (left) the observation function was action-dependent, whereas in our modified version (right) observations only depend on state. The state color for the domain on the right represents the distinct state-dependent observation functions: purple states use the initial observation, while the other states are biased towards either left (blue) or right (red) observations with probability $0.85$.

### H.1 T-maze Details

We use T-maze with corridor length 5 as an instructive example (see Figure 1). The environment has 18 underlying MDP states: one initial state for each reward configuration (reward is either up or down), five for each corridor, one for each junction, and finally one for each terminal state.[11] There are 5 observations in this environment - one for each of the initial states, a corridor observation shared by all corridor states, a junction observation shared by both junction states, and a terminal observation. The action space is defined by movement in the cardinal directions. If the agent tries to move into a wall, it remains in the current state. From the junction state, the agent receives a reward of $+4$ for going north, and $-0.1$ for going south in the first reward configuration. The rewards are flipped for the second configuration. The environment has a discount rate of $\gamma = 0.9$.

---

[11]Technically, we group the four terminal states into a single state for conciseness, which is functionally equivalent.

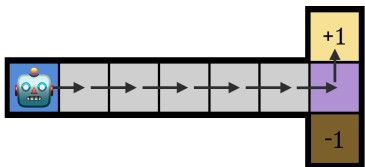 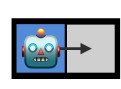 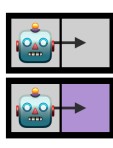 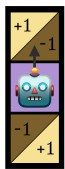

Figure 9: Visualizations of value functions computed using MC (left) and TD (right). MC averages over entire trajectories, so it can associate the blue observation with the upward goal. By contrast, TD computes value by bootstrapping; its value estimates for subsequent observations ignore any prior history.

This environment makes it easy to see the differences between MC and TD approaches to value function estimation. We visualize these differences in Figure 9. MC computes the average value for each observation by averaging the return over all trajectories starting from that observation. By contrast, TD averages over the 1-step observation transition dynamics and rewards, and bootstraps off the value of the next observation. For a policy that always goes directly down the corridor and north at the junction, this leads to an average (undiscounted) return for the blue observation of $+4$ with MC and $(4 - 0.1)/2 = 1.95$ with TD.

## H.2 Analytical Memory-Learning and Policy Improvement Algorithm

Algorithm 1 describes our memory optimization procedure, which reduces the $\lambda$-discrepancy to learn a memory function, then learns an optimal memory-augmented policy.

---

**Algorithm 1** Memory Optimization with Value Improvement

---

**Input:** Randomly initialized policy parameters $\theta_\pi$, where $\Pi = \mathrm{softmax}(\theta_\pi)$, randomly initialized memory parameters $\theta_\mu$, POMDP parameters $\mathcal{P} := (\mathcal{S}, \mathcal{A}, T, R^{\mathcal{S}\mathcal{A}}, \Omega, \Phi, p_0, \gamma)$, number of memory improvement steps $n_{\mathrm{steps},\mu}$, number of policy iteration steps $n_{\mathrm{steps},\pi}$, learning rate $\alpha \in [0, 1]$, and number of initial random policies $n$.
**Output:** Optimized memory-augmented policy parameters $\theta_{\pi_\mu}$ and memory parameters $\theta_\mu$.

*// Initialize random policies and select policy to fix for memory learning.*
$\{\theta_0, \ldots, \theta_{n-1}\} \leftarrow \texttt{randomly\_init\_n\_policies}(n)$
$\theta_\pi \leftarrow \texttt{select\_argmax\_lambda\_discrepancy}(\theta_0, \ldots, \theta_{n-1})$
*// Repeat policy over all memory states.*
$\theta_{\pi_\mu} \leftarrow \texttt{repeat}(\theta_\pi, |\mathcal{M}|)$
*// Improve memory function.*
$\theta_\mu \leftarrow \texttt{memory\_improvement}(\theta_\mu, \theta_{\pi_\mu}, \mathcal{P}, n_{\mathrm{steps},\mu})$
*// Augment POMDP with memory function.*
$\mathcal{P}_\mu \leftarrow \texttt{expand\_over\_memory}(\mathcal{P}, \theta_\mu)$
*// Improve memory-augmented policy over memory-augmented POMDP*
$\theta_{\pi_\mu} \leftarrow \texttt{policy\_improvement}(\theta_{\pi_\mu}, \mathcal{P}_\mu, n_{\mathrm{steps},\pi})$
**return** $\theta_{\pi_\mu}, \theta_\mu$

---

In Algorithm 1, the `policy_improvement` function can be any function which improves a parameterized policy. We use an analytical version of the policy gradient [Sutton et al., 1999] algorithm. The `randomly_init_n_policies` function returns $n$ randomly initialized policies, and the `select_argmax_lambda_discrepancy` function picks the one with the largest $\lambda$-discrepancy. The `memory_improvement` and `expand_over_memory` functions are defined in Appendices H.2.1 and H.3, respectively.

We have noticed that a larger $\lambda$-discrepancy tends to provide a better signal for learning memory functions. Sampling random policies for memory improvement is highly likely to reveal a $\lambda$-discrepancy, as suggested by the theory in Section 3.2. For this reason, we consider many random policies ($n = 100$), then use the policy which had maximum $\lambda$-discrepancy as the basis for memory optimization.

### H.2.1 Memory Improvement Algorithm

In this section we provide pseudocode (Algorithm 2) for the memory-learning algorithm described in Section 4 and used in Algorithm 1. This function takes as input a POMDP $\mathcal{P}$ and memory parameters $\theta_\mu$, and minimizes the $\lambda$-discrepancy of Definition 1. This minimization is achieved through a gradient descent update computed using the auto-differentiation package JAX [Bradbury et al., 2018].

---

**Algorithm 2** Memory Improvement

---

**Input:** Fixed policy parameters $\theta_{\pi_\mu}$, where $\Pi_\mu^{\mathcal{S}_\mathcal{M}}(\mathcal{S}_\mathcal{M} \times \mathcal{S}_\mathcal{M} \times \mathcal{A})$ is the expanded effective policy over state for the policy $\pi_\mu = \mathrm{softmax}(\theta_{\pi_\mu})\,(\mathcal{S}_\mathcal{M} \times \mathcal{A})$, memory parameters $\theta_\mu$, POMDP parameters $\mathcal{P} := (\mathcal{S}, \mathcal{A}, T, R^{\mathcal{S}\mathcal{A}}, \Omega, \Phi, p_0, \gamma)$, number of improvement steps $n_{\mathrm{steps},\mu}$, learning rate $\alpha \in [0, 1]$.
**Output:** Optimized memory parameters $\theta_\mu$.

**for** $i = 0$ **to** $n_{\mathrm{steps},\mu} - 1$ **do**
 *// Augment POMDP with memory parameters $\theta_\mu$*
 $(\mathcal{S}_\mathcal{M}, \mathcal{A}, \Omega_\mathcal{M}, T_\mu, R_\mu^{\mathcal{S}_\mathcal{M}\mathcal{A}}, \Phi_\mathcal{M}, p_{\mathcal{M}_0}, \gamma) \leftarrow \mathtt{expand\_over\_memory}(\mathcal{P}, \theta_\mu)$
 *// Compute TD value function (with memory augmentation)*
 $Q_{\pi_\mu}^0 = W_\mathcal{M}\left(I - \gamma T_\mu \Phi_\mathcal{M} W_\mu^\Pi\right)^{-1}\!:R_\mu^{\mathcal{S}_\mathcal{M}\mathcal{A}}$
 *// Compute MC value function (with memory augmentation)*
 $Q_{\pi_\mu}^1 = W_\mathcal{M}\left(I - \gamma T_\mathcal{M} \Pi_\mu^{\mathcal{S}_\mathcal{M}}\right)^{-1}\!:R_\mu^{\mathcal{S}_\mathcal{M}\mathcal{A}}$
 *// Calculate the $\lambda$-discrepancy*
 $\Lambda_{\pi_\mu} = ||Q_{\pi_\mu}^0 - Q_{\pi_\mu}^1||_{\pi_\mu,2}$
 *// Calculate the gradient of $\Lambda_{\pi_\mu}$ w.r.t. $\theta_\mu$, update memory parameters*
 $\theta_\mu \leftarrow \mathtt{update\_params}(\alpha, \theta_\mu, \nabla_{\theta_\mu}\Lambda_{\pi_\mu})$
**end for**
**return** $\theta_\mu$

---

Here, $\mathtt{update\_params()}$ is any gradient-descent-like update, such as stochastic gradient descent, Adam, etc. As a note, all parameters $\theta$ in these experiments are initialized with a Gaussian distribution, with mean 0 and standard deviation 0.5.

### H.3 Memory Cartesian Product

In this section, we define the memory-Cartesian product function, $\mathtt{expand\_over\_memory()}$, used by Algorithms 1 and 2. This function computes the Cartesian product of the POMDP $\mathcal{P}$ and the memory state space $\mathcal{M}$, as described in Appendix G.1.

---

**Algorithm 3** Memory Cartesian Product ($\mathtt{expand\_over\_memory}$)

---

**Input:** Memory parameters $\theta_\mu$ (with corresponding memory function $\mu$), POMDP parameters $\mathcal{P} := (T, R^{\mathcal{S}\mathcal{A}}, \Phi, p_0, \gamma)$, number of memory states $|\mathcal{M}|$
*// Repeat reward function for each state over each memory $m \in \mathcal{M}$ ($\mathcal{S}_\mathcal{M} \times \mathcal{A}$).*
$R_\mu^{\mathcal{S}_\mathcal{M}\mathcal{A}} \leftarrow \mathtt{repeat\_over\_states}(R^{\mathcal{S}\mathcal{A}}, |\mathcal{M}|)$
*// Calculate transition function memory cross product.*
*// First, calculate the effective memory function over state ($\mathcal{S} \times \mathcal{A} \times \mathcal{M} \times \mathcal{M}$).*
$\mu_\mathcal{S} \leftarrow \mathtt{einsum}('ij, jklm \rightarrow iklm', \Phi, \mu)$
*// Now expand the state transition function to include memory state transitions ($\mathcal{S}_\mathcal{M} \times \mathcal{A} \times \mathcal{S}_\mathcal{M}$).*
$T_\mu \leftarrow \mathtt{einsum}('iljk, lim \rightarrow lijmk', \mu_\mathcal{S}, T)$
*// Calculate observation function memory cross product ($\mathcal{S}_\mathcal{M} \times \Omega_\mathcal{M}$). $I_{|\mathcal{M}|}$ is the identity function over $|\mathcal{M}|$.*
$\Phi_\mathcal{M} \leftarrow \mathtt{kron}(\Phi, I_{|\mathcal{M}|})$
*// Finally, calculate the initial state distribution ($\mathcal{S}_\mathcal{M}$).*
$p_{\mathcal{M}_0} \leftarrow [p_0(s) \text{ if } m = 0 \text{ else } 0 \text{ for } s, m \in \mathcal{S}, \mathcal{M}]$
**return** $\mathcal{P}_\mu = (\mathcal{S}_\mathcal{M}, \mathcal{A}, \Omega_\mathcal{M}, T_\mu, R_\mu^{\mathcal{S}_\mathcal{M}\mathcal{A}}, \Phi_\mathcal{M}, p_{\mathcal{M}_0}, \gamma)$

---

Note that `einsum` is the Einstein summation, and `kron` is the Kronecker product. The augmented initial state distribution is simply the same distribution as $p_0$, except with 0 probability mass over all non-zero memory states, since the memory state always initializes to memory state 0.

## H.4 Closed-Form Memory Learning Experiment Details

For all experiments in Section 4, we run memory optimization on the suite of POMDPs with the following hyperparameters. We optimize memory for $n_{\text{steps},\mu} = 20K$ steps and run policy iteration for $n_{\text{steps},\pi} = 10K$ steps. For all gradient-based experiments, we use the Adam optimizer [Kingma and Ba, 2015].

For the belief-state baselines, solutions were calculated using a POMDP solver from the `pomdp-solve` package [Cassandra, 2003]. The performance of the belief-state optimal policy was calculated by taking the dot product between the initial belief state and the maximal alpha vector for that belief state. This returns a metric comparable to the initial state distribution weighted value function norm, which we use as a performance metric for our memory-augmented agents.

The belief-state solution for the $4 \times 3$ maze was solved using an epsilon parameter of $\epsilon = 0.01$, due to convergence issues with the environment when utilizing the POMDP solver.

## H.5 Closed-Form Experiments on Parity Check

In Section 3.3 we introduced the Parity Check environment as an example of a POMDP with zero $\lambda$-discrepancy for all policies. There we showed that in such environments, we can still detect partial observability by using randomly initialized memory functions. But the question remains: how prevalent are such environments?

We find that these sorts of environments appear to be quite rare. The symmetry disappears if we modify the example even slightly, such as by changing the start state probabilities or by introducing a "stay-in-place" action at any one color. These modifications lead to the $\lambda$-discrepancies in Figure 10, which offer some reassurance that such edge cases are the exception and not the rule.

Besides modifications to the POMDP itself, adding memory can induce a non-zero $\lambda$-discrepancy. In particular, random memory functions reveal a $\lambda$-discrepancy in the system that can be minimized to learn memory, as demonstrated in Figure 4. We now describe this memory optimization procedure.

**Memory Learning**    Memory optimization for the Parity Check experiments in Figure 4 (right) followed almost the same procedure as Algorithm 1 except for selecting the initial policy. In this case, any randomly initialized memoryless policy has zero $\lambda$-discrepancy, but augmenting with a randomly initialized memory function reveals a non-zero measure. Thus, for this experiment, we augment our POMDP with a random memory function *before* choosing the initial policy that maximizes the $\lambda$-discrepancy. All other algorithmic details remained the same.

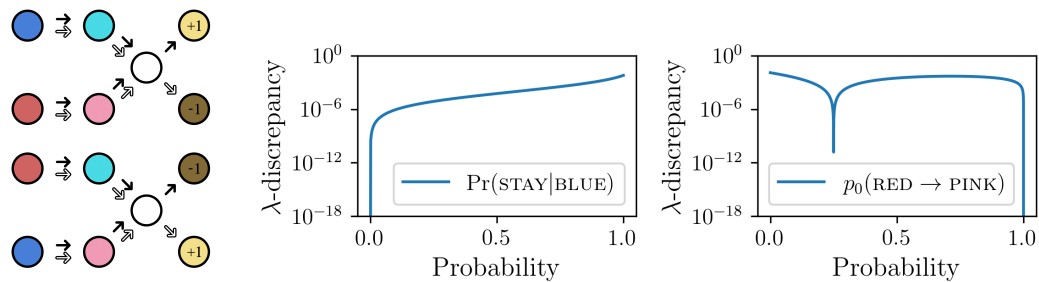

Figure 10: (Left) The Parity Check environment (reproduced from Figure 4). (Right) Minor modifications to the transition dynamics (left) or initial state distribution (right) result in non-zero $\lambda$-discrepancy for almost all policies.

# I  Scaling the $\lambda$-Discrepancy with PPO and RNNs

## I.1  Algorithm Details

We build our $\lambda$-discrepancy minimizing reinforcement learning algorithm on top of online recurrent PPO. Normally, PPO would have two losses: one for the actor and one for the critic.

$$L_{\text{PPO}}(\theta) = \mathbb{E}_\pi \left[ L_{\text{CLIP}}(\theta_{\text{Actor}}, \theta_{\text{RNN}}) - c_V L_V(\theta_{\text{Critic}}, \theta_{\text{RNN}}) + c_{\text{Ent}} H[\pi_\mu(\cdot \mid \omega_t, z_{t-1})] \right]. \quad (11)$$

Here, $L_{\text{CLIP}}$ is the clipped surrogate loss for the actor, defined by:

$$L_{\text{CLIP}}(\theta_{\text{Actor}}, \theta_{\text{RNN}}) = \mathbb{E}_\pi \left[ \min(\rho_t(\theta)\hat{A}_t, \text{clip}(\rho_t(\theta), 1 - \epsilon, 1 + \epsilon)) \right], \quad (12)$$

where $\rho_t(\theta)$ is the ratio between policy probabilities of the old policy and new policy, $\hat{A}_t$ is the estimated advantage at timestep $t$, and $c_V$ and $L_V$ are the value coefficient and value function loss, respectively. $L_V$ is the mean-squared TD error, with targets calculated using a truncated version of $\lambda$-return [Sutton and Barto, 2018]. Finally, $c_{\text{Ent}}$ and $H[\pi_\mu(\cdot \mid \omega_t, z_{t-1})]$ are the entropy coefficient and entropy of the policy $\pi_\mu$ that is parameterized by $\theta_{\text{RNN}}$ and $\theta_{\text{Actor}}$. This term is used for exploration through entropy maximization [Mnih et al., 2016]. For our memoryless baseline, we use a non-recurrent architecture that is a function of only the current observation (and optionally the most recent action).

Our proposed algorithm replaces $L_V$ with $L_{V,\lambda_1,\lambda_2}$, which has three components: two value losses for two separate value heads (each estimating $\lambda$-returns for different $\lambda$s), and our $\lambda$-discrepancy auxiliary loss.

$$L_{V,\lambda_1,\lambda_2}(\theta_{\text{Critic}}, \theta_{\text{RNN}}) = \beta L_\Lambda(\theta) + (1 - \beta)\left( L_{V,\lambda_1}(\theta_{\text{Critic1}}, \theta_{\text{RNN}}) + L_{V,\lambda_2}(\theta_{\text{Critic2}}, \theta_{\text{RNN}}) \right),$$

where $L_\Lambda(\theta)$ is our $\lambda$-discrepancy loss, as defined in Equation 8, and $L_{V,\lambda}$ is the $\lambda$-return-as-target TD error for a given $\lambda$. Here, $\beta \in [0,1]$ is a hyperparameter trading off between the $\lambda$-discrepancy loss and the value losses. Finally, the agent only uses $V_{\lambda_1}$ as its value estimate for its advantage function.

The computational and memory requirements for PPO with and without the $\lambda$-discrepancy auxiliary loss are similar. While the $\lambda$-discrepancy loss requires training a second value head, this only adds two additional layers to the critic network and results in a small number of additional parameters (about 12% overhead). As for computation, the addition of this auxiliary loss only adds a small constant factor to the backpropagation of gradients, since we learn everything end-to-end. Overall, the wall-clock time of runs between the baseline algorithm and our algorithm is comparable.

## I.2  Environment Details

All four of the environments used for evaluation in Section 5 were all re-implementations of environments used in Silver and Veness [2010], in JAX [Bradbury et al., 2018], allowing for massive hardware acceleration. We now give details of these environments.

### I.2.1  Battleship

Partially observable Battleship, a one-player, limited observation variant of the popular two-player board game, was first introduced as a benchmark for partially observable Monte-Carlo planning [Silver and Veness, 2010]. This variant is particularly challenging since the agent has to reason about the unknown positions of the ship and keep track of past shots. The observation space is $\Omega = \{0, 1\}$; at every step, the agent receives a binary observation: 0 if the last shot missed a ship, and 1 if the last shot hit a ship. The state space in this game contains all possible board states, which is astronomically large. The action space is $\mathcal{A} = \{1, \ldots, 10\} \times \{1, \ldots, 10\}$, or all possible grid locations. The agent is only allowed to take valid actions at every step (no position can be selected twice), which is achieved through action masking of the actor. This makes the problem a finite horizon problem, with a horizon at most $10 \times 10 = 100$. The environment terminates when all positions on the grid with a ship are hit. The rewards at every step are -1, with a positive reward of $10 \times 10 = 100$ when all ships are hit/the environment terminates. The discount factor $\gamma$ is set to 1 here, since we are in the finite horizon setting.

At every environment reset, 4 ships of length $(5, 4, 3, 2)$ are uniformly randomly placed on a $10 \times 10$ grid. At every step, to allow for easier learning of all agents, we concatenate the last action to the observation, so that our RNN conditions on the entire history, as opposed to only observations.

### I.2.2 Partially Observable PacMan

Partially observable PacMan (a.k.a. *PocMan*), a variant of the popular arcade game, was also introduced to test partially observable Monte-Carlo planning with a simulator [Silver and Veness, 2010]. In this version, the agent can only observe indicators of its surroundings, including walls in its cardinal directions, whether there is a ghost in its line of sight, a power pellet nearby, or food nearby. These highly obscured agent-centric observations require the agent to localize within the map, seek food and power pellets, and avoid ghosts with only local sensor information.

Concretely, the observation space is an 11-dimensional binary vector. The first four values are 1 if there is a wall in each of the four cardinal directions of the agent. The next element is 1 if there is food Manhattan distance 2 or less away from the agent. The next value is 1 if there is a ghost a Manhattan distance 2 or less away. The next four elements are 1 if there is at least one ghost in the line of sight in each of the four cardinal directions. The last element is if the agent currently has a power pellet.

As with Battleship, the state space in PocMan is also very large. The map is a $19 \times 21$ sized map defined in the original instantiation of the environment [Silver and Veness, 2010], and the state consists of the agent and ghost locations, as well as the binary status of every dot and power pill. The action space has four actions, corresponding to moving in the four cardinal directions. The agent receives a reward of +200 if it eats a ghost, +10 if it collects a pellet, and +20 if it collects a power pellet. The discount factor is set to $\gamma = 0.95$.

The episode terminates when the agent is killed by a ghost, the agent collects all pellets, or the timer for the environment elapses. At the beginning of each episode, all ghost locations are reset, and the agent is placed in the same fixed starting position. In this setting, we also append the agent's most recent action to its observation.

In our work, we implement this domain by adding the observation function on top of the PacMan environment in the Jumanji reinforcement learning framework [Bonnet et al., 2024].

### I.2.3 RockSample

RockSample [Smith and Simmons, 2004], a well-known partially observable benchmark in POMDP planning literature, is a rock-collecting problem simulating a rover scanning potential rock samples and collecting them if they are desirable. RockSample $(n, k)$ is a problem with a grid of size $n \times n$, and $k$ rocks distributed randomly throughout the environment. The two variants we test on have a grid size of $11 \times 11$ and $15 \times 15$, as well as 11 and 15 rocks in each environment, respectively. When instantiated, each RockSample environment for each seed randomly samples rock positions. At the reset of each environment, each of these rocks are uniformly randomly assigned to be either a good or bad rock. The action space in this environment is $4 + 1 + k = 5 + k$. The first 4 actions correspond to moving in the cardinal directions. Action 5 corresponds to sampling in the current position. The last $k$ actions correspond to checking each of the $k$ rocks. Checking a rock will probabilistically tell the agent the correct parity of the rock, depending on the *half-efficiency distance* $d_{\text{hed}}$ and the $l_2$ distance $d_i(s)$ between the agent's position and the rock checked, given the current state $s$:

$$\Pr\left(\text{accurate} \mid s, a = \text{check}_i\right) = \frac{1}{2}\left(1 + 2^{-d_i(s)/d_{\text{hed}}}\right). \tag{13}$$

In this work, we simply set $d_{\text{hed}}$ to be the maximum possible distance between any two points in the grid. Traversing to and then sampling a good rock gives a reward of +10. Sampling a bad rock gives a reward of -10. Exiting to the east border of the environment will terminate the environment and result in a reward of +10. After sampling a good rock, sampling the same rock again will result in a reward of -10. This environment has a discount rate of $\gamma = 0.99$

The state space of this environment is all the possible combinations of rock positions, as well as agent positions and rock parities. The agent receives observations of the form of a vector of size $2n + k$ binary values, where the first $2n$ elements are a two-hot representation of the $xy$-coordinates of the agent. The final $k$ observations are set after choosing a check action, and set a 1 at the position of the checked rock if it appears to be good, and a 0 if it appears to be bad. This observation function is a slight departure from the previous RockSample problem definition [Smith and Simmons, 2004, Silver and Veness, 2010] in two aspects, based on an implementation of RockSample that is used to test RNNs [Tao et al., 2023], as opposed to model-based planning algorithms. The first is that after an agent checks a rock and the agent receives a positive sensor reading, the observation bit corresponding

to this rock is set to 1, and remains 1 until sampled. This makes the memory learning portion of the problem easier, but the agent is still required to remember which rocks it has sampled before, and also deal with the stochasticity of checking. The second change is to not have an explicit negative sensor reading. Instead, after checking a rock, the observation bit corresponding to the sensor reading of the rock remains 0. This makes the problem harder, since the agent has to infer the negative sensor reading, and disambiguate it from a null sensor reading, when the agent takes non-check actions.

### I.3 Discounted Return RNN Results

In the main text (Section 5), we follow the common practice of reporting results for undiscounted returns. However, since our optimization objective considers discounted returns, we also include learning curves for discounted returns in Figure 11. For ease of comparison, we also duplicate the undiscounted learning curves (from Figure 6) in Figure 12. Note that Battleship is unchanged, since that domain is finite-horizon and undiscounted.

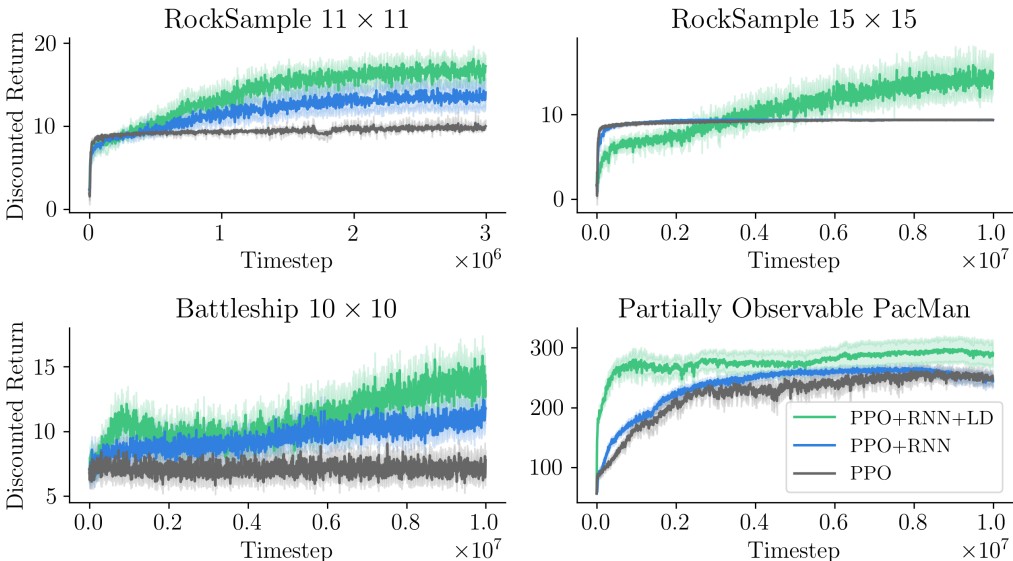

Figure 11: The $\lambda$-discrepancy (LD) performance in discounted returns over recurrent (RNN) and memoryless PPO. Learning curves shown are the mean and 95% confidence interval over 30 runs.

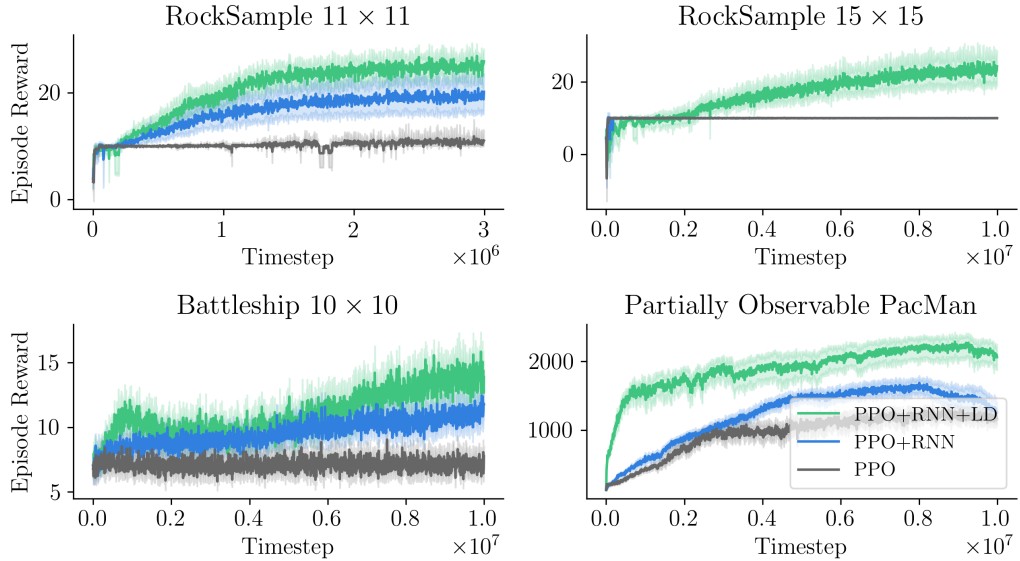

Figure 12: Larger version of Figure 6. The $\lambda$-discrepancy (LD) performance (undiscounted returns) over recurrent (RNN) and memoryless PPO. Learning curves shown are the mean and 95% confidence interval over 30 runs.

## I.4 Experimental and Hyperparameter Details

Our base PPO algorithm is an online PPO algorithm that trains over a vectorized environment, all parallelized using JAX [Bradbury et al., 2018] and the PureJaxRL batch experimentation library [Lu et al., 2022] with hardware acceleration. The hyperparameter sweep was performed on a cluster of NVIDIA 3090 GPUs, and the best seeds presented were run on one GPU for each algorithm, running for 1 to 12 hours, depending on the domain. Our environment is vectorized over 4 copies, and we use truncated-backpropogation through time [Jaeger, 2002] as our gradient descent algorithm, with a truncation length of 128. Our $L_{\text{CLIP}}$ clipping $\epsilon$ is set to 0.2. The value loss coefficient is set to $c_V = 0.5$. We also anneal our learning rate over all training steps, and clip gradients when the norm is larger than 0.5 by their global norm [Pascanu et al., 2013].

We now detail the hyperparameters swept across all environments and all algorithms. We do so in Table 1.

| Hyperparameter | |
| --- | --- |
| Step size | $[2.5 \times 10^{-3}, 2.5 \times 10^{-4}, 2.5 \times 10^{-5}, 2.5 \times 10^{-6}]$ |
| $\lambda_1$ | $[0.1, 0.5, 0.7, 0.9, 0.95]$ |
| $\lambda_2$ ($\lambda$-discrepancy) | $[0.1, 0.5, 0.7, 0.9, 0.95]$ |
| $\beta$ ($\lambda$-discrepancy) | $[0, 0.125, 0.25, 0.5]$ |

Table 1: Hyperparameters swept across all algorithms. Rows labelled with $\lambda$-discrepancy are hyperparameters swept specific to our algorithm.

We conduct this sweep across all environments for 5 seeds each, and use the highest area under the learning curve (AUC) score in each environment to select the best hyperparameters for each environment. Note, we swept over 10 seeds for recurrent PPO and $\lambda$-discrepancy-augmented recurrent PPO for the PocMan environment, due to high variance in returns. After hyperparameter selection, we re-run all algorithms on all environments with the best selected hyperparameters for 30 different seeds to produce the learning curves in Figure 6. We present the best hyperparameters found for both algorithms in Figure 2.

| | Step size | $\lambda_1$ | $\lambda_2$ | $\beta$ |
| --- | --- | --- | --- | --- |
| Battleship | $2.5 \times 10^{-4}$ | 0.1 | 0.95 | 0.5 |
| PocMan | $2.5 \times 10^{-4}$ | 0.95 | 0.5 | 0.5 |
| RockSample $(11, 11)$ | $2.5 \times 10^{-4}$ | 0.1 | 0.95 | 0.5 |
| RockSample $(15, 15)$ | $2.5 \times 10^{-4}$ | 0.1 | 0.5 | 0.25 |

(a) Best hyperparameters found for $\lambda$-discrepancy-augmented recurrent PPO.

| | Step size | $\lambda_1$ |
| --- | --- | --- |
| Battleship | $2.5 \times 10^{-5}$ | 0.7 |
| PocMan | $2.5 \times 10^{-5}$ | 0.7 |
| RockSample $(11, 11)$ | $2.5 \times 10^{-4}$ | 0.7 |
| RockSample $(15, 15)$ | $2.5 \times 10^{-5}$ | 0.5 |

(b) Best hyperparameters found for the recurrent PPO baseline.

| | Step size | $\lambda_1$ |
| --- | --- | --- |
| Battleship | $2.5 \times 10^{-5}$ | 0.1 |
| PocMan | $2.5 \times 10^{-4}$ | 0.7 |
| RockSample $(11, 11)$ | $2.5 \times 10^{-4}$ | 0.1 |
| RockSample $(15, 15)$ | $2.5 \times 10^{-5}$ | 0.1 |

(c) Best hyperparameters found for the memoryless PPO baseline.

Table 2: Best hyperparameters for each environment and each algorithm. Hyperparameters were found using 5 seeds, and taking the maximum AUC.

Our network architectures are standard multi-layer perceptions (MLPs). Both actor and critic networks are two-layer MLPs with ReLU [Nair and Hinton, 2010] activations between layers. The actor network applies a softmax function to its output logits. Our recurrent neural network is a dense layer, followed by ReLU activation, then the GRU cell, then another dense layer. For Battleship, to better condition on the hit-or-miss observation, we add an additional dense layer after the first dense layer of the RNN (which has $2\times$ latent size for this environment), that takes as input the outputs of the previous layer

concatenated with the hit-or-miss bit (a residual connection). For our memoryless baseline, we replace the recurrent neural network with a 3-layer MLP with ReLU activations between each layer. Put together, our actor-critic network takes in an input observation (and optionally, the previous action) and passes it, as well as the previous latent state, through the GRU for a new latent state. For the memoryless baseline, the 3-layer MLP simply encodes the observation (and potentially action) into the latent state, with no recurrence. It uses this new latent state as inputs to both the actor and critic.

We now detail environment-specific hyperparameters in Table 3. All hidden layer latent sizes are the same sizes as the latent sizes. The full implementation of algorithms, experiments, and environments are available at `https://github.com/brownirl/lambda_discrepancy`.

|  | Latent size | $c_{\text{Ent}}$ |
|---|---|---|
| Battleship | 512 | 0.05 |
| PocMan | 512 | 0.05 |
| RockSample $(11, 11)$ | 128 | 0.35 |
| RockSample $(15, 15)$ | 256 | 0.35 |

Table 3: Environment-specific hyperparameters, set across all algorithms. We set the entropy coefficient to a higher value in RockSample because the environment requires more exploration.

### I.5 P.O. PacMan Pellet Probe Visualization Details

We train two RNN hidden state probes in order to generate the memory visualizations in Figure 2. Probes were trained on the hidden state outputs of the RNN + PPO and RNN + PPO + LD agents. Training was done over 2M time steps, where 1M steps were collected from each of these agents. After collection, all trajectories were run through both agents to collect 2M time steps to collect both RNN + PPO hidden states and the RNN + PPO + LD hidden states. With this dataset, the each probe was trained with the corresponding RNN hidden states as inputs, with the pellet occupancy of all potential pellet positions as the target for the prediction. Both probes are 3-layer neural networks with ReLU activations [Nair and Hinton, 2010] between each layer, and a sigmoid function over the final logits to map outputs to 0 and 1. We use a binary cross entropy loss between these predictions and the pellet occupancy targets. The hidden size of the network was 1024, with a step size of 0.0001. At every step of training, a batch of 32 time steps are uniformly randomly sampled from the dataset, and the binary cross entropy loss was minimized. The agent performs 10M steps of training to reach the performance visualized.

### I.6 Small-Scale POMDP Experiments

We also run both recurrent PPO and our $\lambda$-discrepancy-augmented PPO algorithm on the small-scale POMDPs evaluated in the analytical experiments in Section 4. For these experiments, all agents had a latent state size of 32, with $c_{\text{Ent}} = 0.05$, and the same hyperparameters swept as in Appendix I.4. We show results in Figure 14.

These results imply that the baseline algorithm is already sufficient for solving these tasks, and any additional auxiliary losses cannot help with performance, since performance is already near optimal. We would like to note that, as with the results in Figure 6, adding our $\lambda$-discrepancy auxiliary loss never *harms* the performance of PPO.

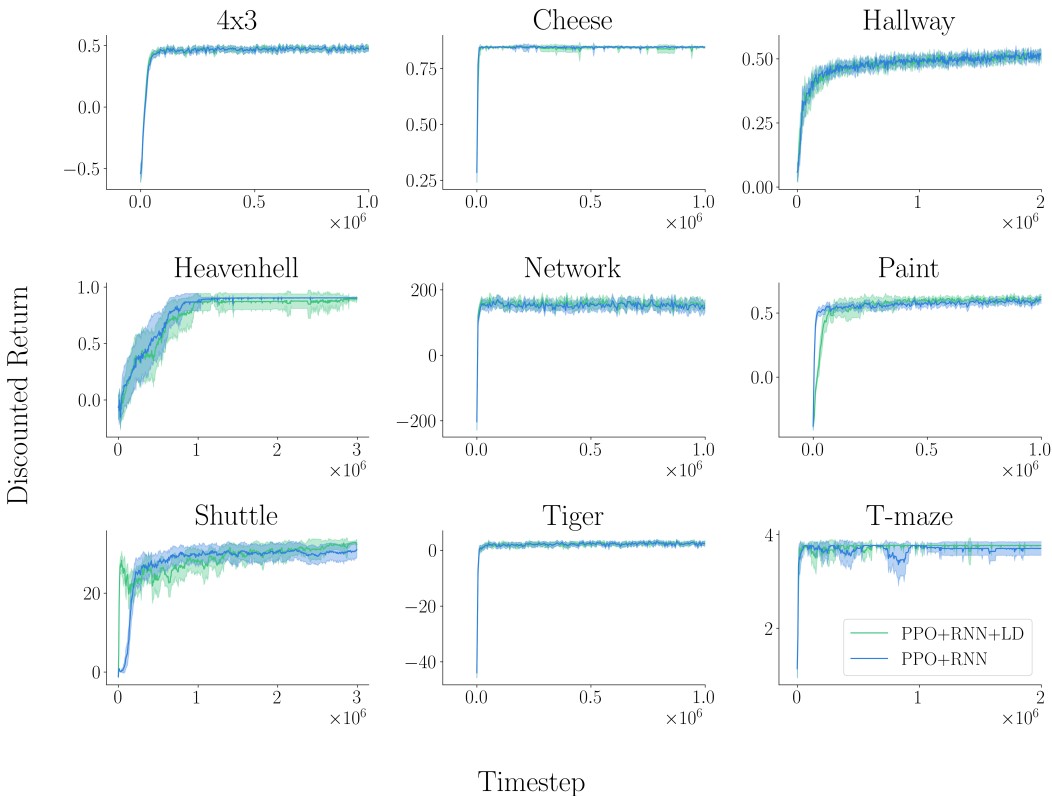

Figure 14: The $\lambda$-discrepancy performance over recurrent PPO on small-scale POMDPs. Learning curves show mean and 95% confidence interval over 30 seeds.

