# OpenReview forum: "Mitigating Partial Observability in Sequential Decision Processes via the Lambda Discrepancy"
_NeurIPS.cc/2024/Conference — NeurIPS 2024 poster_

### Official Review · Reviewer_8yd3 · 2024-07-01

**Soundness:** 3
**Presentation:** 3
**Contribution:** 4
**Rating:** 7
**Confidence:** 4

**Summary:**

The authors propose the lambda discrepancy, a measure of partial observability based on TD-lambda. Importantly, this metric can be estimated from samples of POMDP observations (and a bootstrapped value function estimator) without relying on a transition model, access to POMDP states, or even knowledge of what information the states contain. It is shown theoretically that the lambda discrepancy is zero under a Markovian state representation and almost always non-zero otherwise. Experimentally, it is shown that optimizing the lambda discrepancy can improve representation learning and downstream performance both on smaller POMDP planning domains as well as larger problems solved by PPO.

**Strengths:**

**Originality**: I commend the authors on their vision described in this paper. The lambda discrepancy seems to be widely applicable, scalable, and addresses an important problem broadly relevant to sequence decision making. It leverages well understood techniques in the field (namely, TD-lambda) in a creative new way. To my knowledge, the idea is novel and unique.

**Quality**: The use of the lambda discrepancy as a measure of partial observability is theoretically justified. It is shown that the measure is equal to 0 under a Markovian state representation, while typically being non-zero otherwise. The experiments appear sound and show optimizing the lambda discrepancy aids in learning a good policy.

**Clarity**: See comments below.

**Significance**: I believe this idea has the potential for very high impact. It is general and targets a challenging and important problem.

**Weaknesses:**

While my overall impression of the idea is highly positive, unfortunately I believe that it has not been investigated sufficiently in this work. I don't find the theory/discussion/experiments entirely convincing that the lambda discrepancy is a reliable and broadly applicable metric across general POMDP problems. Furthermore, the manuscript has some serious clarity issues and perhaps tries to convey too much material in too little space (and relegating a lot of important parts to the appendix).

**Quality**: The discussion of some important points -- both theory-wise and experiment-wise -- is missing. In (rougly) chronological order:

(Theoretical:)
- While I appreciate that there is a discussion of when the lambda discrepancy can be 0 aside from when the environment is an MDP (which explores its limitation as a measure of "non-Markovian"-ness), I'm not sure what the conclusion is. Based on Eq 7 and the subsequent discussion, it seems that there are many ways this can occur and no good way of characterizing them. (I couldn't really understand the text nor the example in this section, but if the authors disagree with this point, please bring it up in the rebuttal.)

- There should be some theoretical discussion on how to choose $\lambda_1, \lambda_2$ in the discrepancy. For example, it could be mentioned that for large values of $\lambda \approx 1$, the TD error estimator has very high variance and may not be practical for predicting the TD error.

(Experimental:)
- I would've appreciated some experiments of an "observational" nature to verify the lambda discrepancy actually measures how Markovian the state representations are. For example, maybe you could formulate a class of environments that become more "partially observable" according to a parameter $\epsilon$, and show how the lambda discrepancy changes with $\epsilon$. Another interesting experiment is to train a state representation from the history using an RNN, and compare lambda discrepancy under the state representation with some other known measures of "Markovian"-ness like bisimulation distance [1].

- I appreciate the experiments on larger POMDP planning benchmarks, since they require complex memory that seem non-trivial to learn. However, the transition/observation system is presumably simple enough for someone to specify. It would be interesting to apply the method on domains where the observation/transition function is more complex, such as image-based continuous control.

**Clarity**: Overall, I believe this is the weakest point of the paper. I had a hard time reading the paper since it glossed over important points, with a lot of material placed in the appendix. I understand the struggle to fit everything within the page limit, but there were some paragraphs that I felt did not need to be in the main text as well. A non-exhaustive list of things that could be potentially confusing to a reader:

- The definition of a *Markovian* state representation in text (line 95-97) and mathematically (Eq 1) do not describe the same thing. At the very least, Eq 1 needs to include observations $\omega$.
- The definition of empirical MDPs (lines 128-135) is confusing for a few reasons. First, it's not a real MDP, and the way it's presented makes it seem like an existing concept rather than a new concept by the authors. Second, terms like $Pr(s | \omega)$ are not well defined without further context. I understand that it's somewhat clarified in the next sentence, but I think a precise definition from the start would help someone reading things in order. I suggest it be defined using expectations over trajectories under the policy $\pi$.
- In equation 6, is the introduction of tensor notation necessary? Is it not possible to derive this using probability notation directly? Also, what is the relevance of using Q-functions vs value functions in the definition of lambda discrepancy?
- I cannot make sense of the explanation for when Eq 7 can be zero, even after several reads. It's also unclear to me what about the Parity Check example makes it one of the degenerate cases for Eq 7.

**Significance**: No weaknesses. I believe this work has great potential, but the other weaknesses affect how this work would be received by the broader community.

[1] Zhang, Amy, et al. "Learning Invariant Representations for Reinforcement Learning without Reconstruction." International Conference on Learning Representations.

**Questions:**

Please see weaknesses above.

**Limitations:**

Yes.

---

> ### Author Rebuttal · Authors · 2024-08-07
>
> Thank you for the kind words and feedback. We're glad you had such a positive impression of the idea and see potential for very high impact. You cited clarity as your biggest concern, so we'll address that here and put our other comments in the global response.
>
> **Markovian Representation**
>
> We apologize for the confusion. There was a typo in line 97. It should instead read "next representation z_{t+1}". The definition can be applied to states (s), observations (ω), and even memory-augmented observations (ω, m) by replacing z with the relevant quantity. If equation 1 holds, the corresponding representation is Markovian.
>
> **Empirical MDP**
>
> You're correct that Pr(s|ω) is an expectation over trajectories under the policy. Assuming all observations have non-zero probability, we can write this in terms of trajectories $\tau=(s_0,ω_0,a_0,r_0, s_1,ω_1,a_1,r_1,...)$:
>
> $$Pr(s|ω)=\frac{𝔼\_{\tau\sim π}[\sum_{t=0}^∞ γ^t𝟙[s_t=s,ω_t=ω]]}{𝔼\_{\tau\sim π}[\sum_{t=0}^∞ γ^t 𝟙[ω_t=ω]]}.$$
>
> This is equivalent to the definition we use in Appendix C, but we agree it is easier to understand it this way up front.
>
> Your point that the empirical MDP is “not a real MDP” is interesting. MDPs are simply models of environments and can be of arbitrary quality. The empirical MDP is one such model. It is the one practitioners use implicitly whenever they treat POMDP observations as Markovian "states". The empirical MDP model consists of (Markovian) transition and reward functions that condition on observations, which we define in terms of Pr(s|ω), and which, crucially, might make different predictions from the POMDP model. We can thus view Pr(s|ω) as a weighting scheme for aggregating the behavior of the states that produce each observation.
>
> Model selection is not a new idea, but our specific characterization of the empirical MDP for a given POMDP is novel to the best of our knowledge.
>
> **Tensor Notation and Q vs. V**
>
> Defining LD using probability notation directly would be prohibitively cumbersome and a poor use of space (see Appendix C). Tensor notation succinctly expresses the concept and has precedent in many papers [Lagoudakis & Parr 2003, Tagorti & Scherrer 2015, Gehring et al. 2016]. Q-values are easy to express with tensors that track action and observation indices separately, and they are less likely than V-values to succumb to Parity Check symmetry, since they avoid an additional summation over actions.
>
> ### LD=0 cases and Parity Check
>
> We found a new way to characterize these edge cases that streamlines the section, removes the need for Eq. 7 entirely, and that we hope is much clearer.
>
> We start with Defn. 1:
> $$ Λ_π^{λ_1,λ_2} := \|\| Q_π^{λ_1} - Q_π^{λ_2}\|\| = \Big\|\Big\|W\Big(I-γTK_π^{λ_1})^{-1}-(I-γTK_π^{λ_2})^{-1}\Big):R^{SA}\Big\|\Big\|,$$
> where $K^λ_π := (λ Π^S + (1-λ) \Phi W^Π)$, and the `:` operator denotes a tensor product over two indices.
>
> Let us parse this equation by considering one of the two Q-functions in isolation (say $λ_1$).
>
> **Policy Tensors.** First, note that $Π^S$ and $\Phi W^Π$ are mappings from states to distributions over state-action pairs $s\mapsto\Delta(\tilde s,a)$. We call the former the *MC policy tensor*, ${Π^S}\_{s,\tilde s,a}=𝟙[s=\tilde s]\sum\_ω \Phi\_{s,ω}π\_{ω,a}$, which maps each state the expected policy under its observation distribution. We call the latter the *TD policy tensor*, $(\Phi W^Π)_{s,\tilde s,a}=\sum\_ω \Phi\_{s,ω} π\_{ω,a} W\_{ω,\tilde s}$, which spreads the policy probabilities for a given observation $ω$ across all states $\tilde s$ that produce $ω$, weighted by $W=Pr(s|ω)$. $K\_π^{λ_1}$ is a convex combination of these policy tensors, parameterized by ${λ_1}$.
>
> **Transition-policy tensors.** Left-multiplying a policy tensor by $T$ produces an $(S×A×S×A)$ transition-policy tensor, which describes the probability of each state-action transition $(s,a)\mapsto\Delta(s',a')$ under the policy. Intuitively, TD($λ_1$) computes Q-values for a POMDP as though state-action pairs are evolving according to $TK\_π^{λ_1} = λ_1 TΠ^S + (1-λ_1)T\Phi W^Π$, which is a mixture of the policy dynamics under two models: the MC transition-policy ($TΠ^S$) and the TD transition-policy ($T\Phi W^Π$).
>
> **Final projections.** The expression $(I - γTK\_π^{λ_1})^{-1}$ computes the state-action successor representation for this hybrid transition model, then multiplication by $R^{SA}$ computes state-space $Q$-values. Finally, these are projected through $W$ to compute observation-space $Q$-values.
>
> Returning to the question of when $Λ=0$ for all policies, there are 3 cases:
>
> 1. Policy tensors are equal: $K_{π}^{λ_1}=K_{π}^{λ_2}$. This simplifies to $Π^S=\Phi W^Π$, which is satisfied if and only if $\Phi W = I$, i.e. each observation is generated by exactly one hidden state. This means the POMDP is actually a block MDP.
>
> 2. Transition-policy tensors are equal: $TK_{π}^{λ_1}=TK_{π}^{λ_2}$. In this case, the POMDP is equivalent to an MDP. Both transition-policy tensors lead to the same state-action successor representation, so any reward function will lead to the same value under both models.
>
> 3. Any differences between transition-policy tensors (and thus the state-action successor representation) are subsequently projected away by either $W$, $R^{SA}$, or both. In this case, LD=0, but the effective MDP may be lossy. Fortunately, such pathological environments appear to be quite rare, and they are easy to handle, as we discuss below.
>
> Parity Check is an example of **case 3**. One can check (by hand or using our code) that the $λ_1$ and $λ_2$ transition-policies (and thus the successor representations) differ with and without aliasing. However, these differences are perfectly symmetric with respect to observations and rewards, so they are projected away by $W$ and $R^{SA}$. The Q-values are therefore zero at every observation, and so the LD=0 for all policies. Nevertheless, as we cover in our global response, we can still detect and mitigate partial observability for this problem.

---

> > ### Comment · Reviewer_8yd3 · 2024-08-12
> >
> > Thank you for your clarifications. Overall, I found the authors' rebuttal to be both substantial and effective at addressing the majority of my concerns.
> >
> > - The authors have clarified my concern re: pathological cases where LD = 0 in POMDPs. The additional experiments showing how it can be practically prevented is great. These pathological cases are reminiscent of saddle points -- theoretically an issue for gradient descent but not a significant concern in practice.
> >
> > - The authors have promised to include practical insights of choosing $\lambda_1, \lambda_2$ when computing the LD, which I think will be valuable.
> >
> > - Per my suggestion, the authors have included an experiment interpolating between MDPs and POMDPs and verifying that LD increases. It's on a toy domain but this is a positive -- it's illustrative, easy to understand, and insightful.
> >
> > A couple of concerns still stand:
> >
> > - I still believe the work would be significantly improved by experimenting on more complex POMDPs. This would alleviate concerns about scalability, e.g. TD-$\lambda$ may not be so easy to accurately estimate for high $\lambda$. In fact, despite its popularity in deep RL, there has been evidence that similar estimates like GAE-$\lambda$ do not need to be that accurate for deep RL to work well [1].
> >
> > - Re: "empirical MDPs", I'm skeptical that the transition function $T(w' | w, a)$ is actually Markovian. Can't we get a better prediction of $w'$ given not just $w$ but the entire sequence of observations $w_1, \ldots, w_t$ up to the current timestep?
> >
> > In any case, I think the work has significantly improved and I'm pleased to recommend acceptance. Congratulations to the authors on a great idea.
> >
> > [1] Ilyas, Andrew, et al. "A Closer Look at Deep Policy Gradients." International Conference on Learning Representations.

---

> > > ### Author Response · Authors · 2024-08-14
> > >
> > > > Re: "empirical MDPs", I'm skeptical that the transition function $T(ω'|a,ω)$ is actually Markovian. Can't we get a better prediction of $ω'$ given not just $ω$ but the entire sequence of observations $ω_1, …, ω_t$ up to the current timestep?
> > >
> > > Great question. Yes we can, and that's precisely the point. If we want to predict the *actual*, true-to-the-environment next observation, we are absolutely better off conditioning on the entire history. If we view $T(ω'|a,ω)$ as a marginal over the transition dynamics for all POMDP trajectories leading to ω, then this quantity is indeed non-Markovian, non-stationary, and policy-dependent.
> > >
> > > At the same time, we are also free to invent a stochastic tensor of transition probabilities and pretend that is an accurate model of the environment. If we do, accurately predicting the imaginary next state in an MDP governed by this stochastic transition tensor requires no more than a single state and action (by definition). It's just that the predicted MDP state may bear no resemblance to the *actual* next observation. If the probabilities in this transition tensor are those of the empirical MDP (as we define it), then this is exactly what TD(0) is doing implicitly when we apply it to the observations of the POMDP.
> > >
> > > So whether $T(ω'|a,ω)$ is Markovian depends on whether we are viewing it as a marginal distribution for the POMDP, or a transition function for a hypothetical MDP.
> > >
> > > We hope this clarifies things!

---

### Official Review · Reviewer_AeCz · 2024-07-10

**Soundness:** 3
**Presentation:** 3
**Contribution:** 2
**Rating:** 6
**Confidence:** 3

**Summary:**

This paper studies policy planning in a partially observable Markov decision process (POMDP). It has been acknowledged that planning in a POMDP is computationally challenging in general. This paper attempts to address this challenge by proposing novel planning heuristics First, the authors formalize the $\lambda$-discrepency, which measures the distance of $\lambda$-returns using different $\lambda$ parameters. The learner then obtains a policy by minimizing the $\lambda$-discrepency. Simulations results demonstrate that the proposed method could outperform standard PPO using recurrent NN in a select set of POMDP environments.

**Strengths:**

- The idea is novel and reasonable. There is a natural connection between $\lambda$-return and the Markov property of the underlying environment. By minimizing the $\lambda$-discrepency using RNN policies, the learner is able to extrapolate a state representation that can sufficiently summarize the past history.
- The concept of $\lambda$-discrepency is novel and could be useful for RL researchers working on other policy learning problems.
- The authors demonstrate the performance improvement on a select set of complex POMDP environments. This suggests that the proposed approach could impact some practical domains.

**Weaknesses:**

- As for Theorems 1 and 2, do the policies $\pi$ have to be Markov policies only taking the current state $S$ as input? If so, the underlying environment is expected to be a block MDP. I am wondering if the result could be more general if we allow non-Markov policies with memory. Could the authors make the connection with the weakly revealing POMDPs?
- The authors could further elaborate on the training procedures for minimizing the loss function in Eq. 8. Please see my questions in the next section.
- Additional discussion on why minimizing $\lambda$-discrepency could improve the planning performance would be appreciated. Could the authors provide some intuitions?

**Questions:**

- What does LD stand for in Figure 4?
- How to select parameters $\lambda_1$ and $\lambda_2$ when minimizing the loss function in Eq. 8? Does one have to try multiple pairs of such $\lambda$s?
- For policies $\pi$ in Theorems 1 and 2, are they Markov policies only taking the current state $S$ as input? If so, it excludes the non-Markov policy, which could have memories.

**Limitations:**

The authors have adequately addressed the limitations of the paper.

---

> ### Author Rebuttal · Authors · 2024-08-07
>
> Thank you for the feedback and questions.
>
> We will group two of your comments together, and respond to the remainder in order.
>
> > *As for Theorems 1 and 2, do the policies π have to be Markov policies only taking the current state S as input? If so, the underlying environment is expected to be a block MDP. I am wondering if the result could be more general if we allow non-Markov policies with memory.*
>
> > *Q3: For policies π in Theorems 1 and 2, are they Markov policies only taking the current state $S$ as input? If so, it excludes the non-Markov policy, which could have memories.*
>
> There seems to be a minor misconception. The agent’s policy $π(a|ω)$ is conditioned on observations---not hidden state. Rather, the *state policy tensor*, $Π^S\_{s,s',a}=𝟙[s=s']\sum\_ω \Phi\_{s,ω}π\_{ω,a}$, is the *effective* policy over states; this tensor maps states to the expected policy under their observation distribution. At no point do we require the agent’s behavior to be conditioned on unobservable quantities. However, our definition does also support policies that condition on both the observation and the memory state, via the memory augmentation process we outline in Sec. 4.
>
> ---
>
> > *Could the authors make the connection with the weakly revealing POMDPs?*
>
> Weakly revealing POMDPs (Liu, Chung, et al. 2022) are an interesting class of POMDPs in which an $m$-step future contains enough information to reveal the current state, given sufficient samples, and the paper authors argue that this condition makes partial observability “not scary”. At a high level, the LD also helps to make partial observability “not scary”, by providing a way to detect and mitigate it. So it would make a lot of sense if the weakly revealing literature ultimately turns out to have some connection to this work. We have not been able to prove a formal connection yet, but if you have suggestions on how to do this, please let us know! Ultimately, we hope that the LD will be a useful tool for analyzing many aspects of partially observable reinforcement learning.
>
> ---
>
> > *Additional discussion on why minimizing λ-discrepency could improve the planning performance would be appreciated. Could the authors provide some intuitions?*
>
> Certainly! Lower LD leads to state representations that are more predictive of the future. Better representations lead to better value function estimation, and this leads to better performance. Why does it help to minimize this LD quantity specifically? Well, to the extent that optimal behavior requires knowledge of the underlying hidden state, zeroing the LD is a reliable way to ensure that the agent’s representation contains a complete picture of that hidden state. On the other hand, if optimal behavior only requires conditioning on the current observation, then LD minimization won’t improve performance, but it can still provide additional confidence that the current observation is indeed sufficient.
>
> To see what these “better” representations look like, check out “Memory visualization” in our global response above. We also find that LD has a minimum at zero and increases with partial observability (see ‘Varying “Markovness”’). We also conjecture that reducing LD also improves value estimation error. We have not yet proved a formal link, but this seems plausible, particularly given that LD is defined as the difference between Q-values, which is essentially a type of value error. Moreover, the empirical results in our paper suggest we are indeed learning better representations, which should allow better value estimation.
>
> ---
>
> ### Questions
>
> > *Q1: What does LD stand for in Fig. 4?*
>
> Sorry for the confusion. LD stands for the Lambda Discrepancy in this Fig. 4. We’ve updated the caption to clarify this.
>
> ---
>
> > *Q2: How to select parameters λ_1 and λ_2 when minimizing the loss function in Eq. 8? Does one have to try multiple pairs of such λ s?*
>
> Thanks for the question. Please see “Understanding and selecting λ parameters” in our global response.
>
> ---
>
> > *Q3.*
>
> (Addressed above.)

---

> > ### Comment · Reviewer_AeCz · 2024-08-12
> >
> > I thank the authors for their detailed response. They have addressed my questions. Overall, this paper proposes novel algorithms for policy learning in MDPs with hidden confounders, and will have an impact on the AI community. On the other hand, I also agree with Reviewer rpiT regarding the comparison with other SOTA POMDP algorithms. Due to these reasons, I will keep my current score.

---

### Official Review · Reviewer_rpiT · 2024-07-13

**Soundness:** 2
**Presentation:** 3
**Contribution:** 2
**Rating:** 5
**Confidence:** 3

**Summary:**

The authors consider a problem of reinforcement learning in partially observable systems. Instead of constructing an information state using the entire trajectory history with some peripheral objective, the authors propose using discrepancy in the values of TD$(\lambda)$ estimates for different values of $\lambda$. Their argument is that TD(0) makes a Markovian assumption, whereas TD(1) (Monte Carlo) does not, therefore any discrepancy in their values can be used to construct a metric called the $\lambda$-discrepancy, which measures the non-Markovian nature or partial observability of the currently used state representation. The authors then use this $\lambda$-discrepancy to construct an RL algorithm that learns a memory function to mitigate partial observability. They do this by traiing two different recurrent networks to predict TD$(\lambda)$-values fpr two different values of $\lambda$ and add an auxiliary loss that minimizes the discrepancy between these twp. Using numerical experiments, they show that their method performs better or at least as good as conventional history based methods for POMDPs.

**Strengths:**

The paper is well written and the claims and explanations are clearly given. The new $\lambda$-discrepancy metric proposed by the authors is an interesting one and the entire approach is novel and useful.

**Weaknesses:**

The metric defined by the authors requires operating over the entire state, observation and action space for its computation as given by the authors in their theoretical derivation. The authors do not explain how the per state/observation, action value determined in practice relates to this metric?

**Questions:**

1. The authors state in Line 146 that: "Without state, a model-based comparison would require variable-length history inputs to ensure a Markovian representation, and the computational and memory requirements would scale exponentially with history length. Instead, we propose a model-free approach, using value functions defined only over observations, to approximate this comparison in a tractable way." However, the authors also use two RNNs in their proposed approach that uses the entire history. So how does this argument hold?
2. The authors have given one example (Parity check environment) where equation (7) is satisfied and the $\lambda$-discrepancy metric fails. The authors also show that this can be addressed by perturbing the initial state distribution or action space. How can such examples be identified without knowledge of the underlying model and how can they be practically addressed as the changes suggested here are changes to the environment which the agent cannot make.
3. How will the convergence analysis of PPO change after the addition of the auxiliary loss given by equation (8)? The authors have proven that $\lambda$-discrepancy reflects partial observability only when the corresponding Q functions are learnt. When these functions are being learnt (still have errors) the $\lambda$-discrepancy will also have an error. How do the authors address this?
4. The authors compare their approach with a recurrent PPO baseline. Why is there no comparison with state of the art POMDP methods for these problems?
5. The proposed algorithm requires two Q networks, thereby doubling the compute requirements of the critic in actor-critic methods. Is the gain in performance sufficient to justify this?
6. How does the memory learnt by this approach compare with representations learnt by POMDP based approaches such as PSRs etc.?
7. Can $\lambda$-discrepancy metric be defined at a state/observation or state/observation x action level? If not, it is not clear how the quantity given in equation (8) relates to the one in Definition 1?

**Limitations:**

The authors address the limitations of their work in a separate section in the Appendix.

---

> ### Author Rebuttal · Authors · 2024-08-07
>
> Thank you for your review and feedback. It seems that your main concern is the misconception that LD “requires operating over the entire state, observation and action space for its computation.” We apologize if this was confusing. While the *derivation* of the metric uses tensors that operate over the entire state, action and observation space, the actual estimation and minimization of the metric only uses estimates of TD($λ_1$) and TD($λ_2$), which are computed directly from sampled (observation, action) pairs. This means that our method does not require the underlying model of the POMDP to mitigate partial observability, only samples.
>
> We realize this may not have been clear in our submission, so we’ve added the following discussion near the end of Sec. 4:
>
> > Despite its definition in terms of $S$, we can express the $λ$-discrepancy in a form amenable to estimation from observable quantities only. By selecting the appropriate weighted norm, the $λ$-discrepancy becomes an expectation over observation-action pairs (or, in the case of memory, observation-memory-action tuples):
> $$Λ\_{\mathcal{P},π}^{λ\_1,λ\_2} := \|\|Q\_π^{λ\_1} - Q\_π^{λ\_2}\|\|\_π = 𝔼\_{π}\left[ Q\_π^{λ\_1}(ω, a) - Q\_π^{λ\_2}(ω, a) \right]$$
> The norm is taken over the on-policy joint observation-action distribution $Pr(ω)π(a|ω)$, which allows us to estimate the metric using samples generated by the agent’s interaction with the environment. We show how to use it as an optimization objective in the following section.
>
> We hope by clarifying our objective's dependence on only observables we've highlighted the broad applicability of this method in partially observable domains.
>
> ---
>
> We will now respond to your questions.
>
> 1. *However, the authors also use two RNNs in their proposed approach that uses the entire history. So how does this argument hold?*
>
>     Sorry for the confusion. We meant to say “Without state, a model-based comparison would require variable-length history inputs **per time step** to ensure a Markovian representation”. So the input size for this model-based comparison would be on the order of O(t), whereas the input size to an RNN is fixed and always O(1). To clarify, we only use a single RNN, with two value *heads* conditioned on the same internal representation (new Fig 2, left).
>
> 2. *How can such examples be identified without knowledge of the underlying model and how can they be practically addressed as the changes suggested here are changes to the environment which the agent cannot make.*
>
>     Good question! We address this with a new experiment in the global response (“Parity Check”).
>
> 3. *How will the convergence analysis of PPO change after the addition of the auxiliary loss given by Eq. (8)? When these functions are being learnt (still have errors) the λ-discrepancy will also have an error. How do the authors address this?*
>
>     While we are unaware of any general convergence guarantees for PPO (much less recurrent PPO), our empirical results show that the LD loss frequently performs better (and never performs worse) than the RNN + PPO baseline.
>
>     You are correct that the value functions will be inaccurate during learning, and so too will the LD. Nevertheless, just as value functions are commonly useful for RL even before they have fully converged, so too is the LD, as our experiments demonstrate.
>
> 4. *The authors compare their approach with a recurrent PPO baseline. Why is there no comparison with state of the art POMDP methods for these problems?*
>
>     Our approach is introducing a new metric, the LD, as opposed to advocating for a particular algorithm like PPO. The LD is not limited to only PPO, but could be leveraged in essentially any RL method that uses value functions. We are considering extensions to other approaches for future work, where we believe the LD still has great potential to improve existing algorithms.
>
>     We aren’t sure what you mean by state-of-the-art POMDP methods. If you mean belief-state methods like POMCP, it is unclear what such a comparison would reveal, since those methods assume knowledge of the transition, reward, and observation functions, and the set of hidden states. Our method only assumes access to on-policy observations while acting in the environment, which is a much more challenging setting.
>
> 5. *The proposed algorithm requires two Q networks, thereby doubling the compute requirements of the critic in actor-critic methods. Is the gain in performance sufficient to justify this?*
>
>     The proposed algorithm requires two value function *heads* (3 layer MLPs), each conditioned on the hidden state of a single RNN. The extra computational overhead is only for this additional value head---about a 12% increase in parameters vs. the RNN baseline, as we discuss in Appendix I. We feel this is a reasonable tradeoff, given the performance gains.
>
> 6. *How does the memory learnt by this approach compare with representations learnt by POMDP based approaches such as PSRs etc.?*
>
>     Good question! We added a new experiment in our global response visualizing these memory functions for the methods in Sec. 5.
>
>     Our approach uses memory functions to incrementally update a summary of past observations. By contrast, PSRs use tests of potential future trajectories, where the main challenge lies in finding these tests. In prior work [Singh, Littman et al., 2003], PSRs were able to learn representations for domains as large as 58 states and 21 observations. By contrast, Battleship (without accounting for possible ship locations) has ~2^100 states. PacMan, even when ignoring food dot configurations, has ~5^400 states. To the best of our knowledge, PSRs have not yet been scaled up to domains of the size we study here.
>
> 7. *Can λ-discrepancy metric be defined at a state/observation or state/observation x action level? If not, it is not clear how the quantity given in Eq. (8) relates to the one in Definition 1?*
>
>     We hope our initial clarification above answered this question!

---

> ### Comment · Reviewer_rpiT · 2024-08-12
>
> I thanks the authors for their explanations. Based on these, I have decided to increase my score. Below are my responses to your explanations:
> 1. Here is one paper that provides convergence analysis for policy gradient algorithms: https://www.jmlr.org/papers/volume22/19-736/19-736.pdf . My question was regarding the impact of your auxiliary loss in such analyses.
> 2. By state of the art POMDP methods, I mean methods such as predictive state representations, deep recurrent Q learning etc., where the algorithm is designed primarily to handle partial observability. Instead of predicting tests as in PSR, there are also auxiliary tasks such as predicting next observation which have been shown to be useful for information state learning in the literature.

---

### Official Review · Reviewer_pbjp · 2024-07-19

**Soundness:** 3
**Presentation:** 3
**Contribution:** 4
**Rating:** 7
**Confidence:** 3

**Summary:**

The Authors address the problem of missingness or partial observability in Markov Decision Problems (MDPs), where the Markov assumption does always hold. They introduce a measure, the Lambda Discrepancy (LD), which serves as an indicator for how closely a Markovian state representation is given.

The LD learns two different value estimates, ideally one making an implicit Markov assumption and one that does not. It is also shown that using the LD it is often clear whether a problem can be defined with a Markovian state representation or not.

Further, the authors use the LD to define an auxiliary loss and included it into a Proximal Policy Optimization algorithm. By using two value estimators for, one for each of two distinct lambdas, the resulting agent outperformed agents that were trained without LD on multiple tasks.

**Strengths:**

- The paper bridges nicely from theory to empirical experiments. The readers are first taught the theory and origin of the Lambda Discrepancy. Then the paper explains the possibility to use LD to detect partial observability before extending this possibility by integrating the discrepancy into the loss function of a policy optimizing algorithm. -> Great story telling.

- The LD method is versatile and can be used for any optimization algorithm using value estimation. Hence, due to its effectiveness and limited added complexity to an optimization algorithm, the Lambda Discrepancy is an effective way to improve existing optimization algorithms.

- The experimental results are used in a concise and convincing manner. The language is clear and to the point.

**Weaknesses:**

- In section “Related Work”, it is said that the “dominant paradigm for resolving partial observability are memory-based approaches”. The experiments also use a recurrent PPO and compare it to recurrent PPO with lambda discrepancy but it is not explained how they relate to other memory-based methods.

- The paper shows experimental results for POMDPs, but there is no example of an environment that is a pure MDP for comparison. Ideally, it would be most interesting seeing results for environments with Markovian state representation that gradually shift to POMDPs that are not Markovian.

- The “Limitations” section should be part of the main corpus.

- Block MDPs are mentioned as a possibility for the LD to be zero, but their relevancy - whether they remain to be an edge case - is not discussed.

- The paper explains how to interpret the extremes of $\lambda$ being 0 or 1. However the explanation of “Intermediate values of λ smoothly interpolate between the two” ([pdf](zotero://open-pdf/library/items/7RNSWK98?page=5&annotation=ESN6JY42)) (l. 163) values is not intuitive. An example at $\lambda=0.5$ would be helpful, where not only the previous state is taken into account (as would be with $\lambda=0$ in an MDP setting) but neither all of the previous observation (as would be $\lambda=1$), but a balance in between.

- While it is stated that the “approach scales to challenging partially observable domains” the experiments remain on a low complexity levels and no comment is made about how the scaling the complexity of environments affects the performance using the LD method.

**Questions:**

- Q1: Could you please clarify the setup regarding the $\text{L}_\Lambda$ loss. You train two value function models for two distinct $\lambda$ values in parallel? Hence, the models share a single loss function? This means after training is complete we end up with two models. How are the results of both networks used/combined?

- Q2: How does the LD method benefit when there is existing domain knowledge about an environment? How would knowledge whether the environment should be modeled as a (PO)MDP including how many prev. states are required to describe the next state be taken into account? In that case, what are the expected advantages of using the LD method?

- Q3: How is the LD method affected when switching between online and offline-RL settings?

**Limitations:**

- Using LD for more complex environments results in adding another value estimator to the optimization algorithm. For complex environments requiring large models this can become a limiting factor.

- Introducing the LD loss introduces multiple hyperparameters $\lambda_1, \lambda_2$ and $\beta$. While it is clear from empirical research that the $\lambda$s should be on separate ends of the interval $[0,1]$ , it is less so for $\beta$. Performing hyperparameter optimization for these three additional parameters is a further addition to computational requirements using LD.

---

> ### Author Rebuttal · Authors · 2024-08-07
>
> Thank you for your response and feedback! We are glad to hear you thought the writing was “clear and to the point.”
>
> We will respond to your comments in order.
>
> ---
>
> > *The experiments also use a recurrent PPO and compare it to recurrent PPO with lambda discrepancy but it is not explained how they relate to other memory-based methods.*
>
> Thank you for the suggestion. While we use RNNs (specifically GRUs) in our experiments due to their popularity in RL, there are other architectures we have not considered. Transformer-based architectures have recently been studied in reinforcement learning settings, and have been shown to help in long-term memory tasks [1]. Seeing as our metric poses no constraints on function approximation or network architecture, an interesting direction for future work would be to see if the LD objective would benefit Transformer-based architectures as well.
>
> As for methods that explicitly consider partial observability with memory, the most dominant approaches have used belief-states. Belief-state methods [2] are a class of solution methods for POMDPs that have seen substantial success in robotics [3], but most approaches are intractable for even small environments where the dynamics are unknown, since most methods rely on some form of state information to form the belief state that acts as memory.
>
> - [1] Ni, Ma, Eysenbach, Bacon. NeurIPS, 2023.
> - [2] Kaelbling, Littman, Cassandra. Artificial Intelligence, 1998.
> - [3] Thrun, Burgard, Fox. Probabilistic Robotics. MIT Press, 2005.
>
> ---
>
> > *Ideally, it would be most interesting seeing results for environments with Markovian state representation that gradually shift to POMDPs that are not Markovian.*
>
> Thanks for the suggestion. We ran this experiment for T-Maze and included the results in our global response.
>
> ---
>
> > *The “Limitations” section should be part of the main corpus.*
>
> We had originally moved the limitations section to the appendix due to space constraints, but we will move this section back into the main corpus with the extra page allotted for the camera-ready.
>
> ---
>
> > *Block MDPs are mentioned as a possibility for the LD to be zero, but their relevancy - whether they remain to be an edge case - is not discussed.*
>
> We proved block MDPs have LD=0, and this is good, because block MDPs are fully observable; they are MDPs. In fact, classic MDPs can be viewed as block MDPs with a single observation per state. Block MDPs are therefore not an edge case, but an indication that our metric is correctly labeling MDPs as having zero partial observability.
>
> ---
>
> > *However the explanation of “Intermediate values of λ smoothly interpolate between the two” (pdf) (l. 163) values is not intuitive. An example at would be helpful*
>
> We apologize for the confusion. You are right that TD(λ=0) and TD(λ=1) don’t provide much intuition on how the intermediate lambda values will behave. For intermediate λ values, TD(λ) is a weighted average of all the n-step returns. The λ parameter controls the weighting scheme: lower values of λ give more weight to the shorter rollouts. When λ=0.5, this defines a weighting scheme where each n-step rollout contributes half as much to the weighted sum as the (n-1)-step rollout. It is possible to view TD(λ) as operating in either a backwards-looking or forwards-looking way, but in the paper, we always take the forward view: the λ parameter does not determine whether previous states are taken into account, but rather how much each n-step future return contributes to the weighted average.
>
> ---
>
> > *The experiments remain on a low complexity levels*
>
> Thanks for the feedback. Please see “Environment Complexity" in our global response above.
>
> ---
>
> ### Questions
> 1. *Could you please clarify the setup regarding the LD loss?*
>
> We have provided a visualization of the network architecture in new Fig. 2 (left). We train two value heads and a policy head on top of a single shared recurrent representation. Only one of the value heads is needed for the PPO objective, but both are used in computing the LD loss, and the recurrent representation is trained end-to-end. For more details, see Appendix I. We will add the figure to the camera-ready if the paper is accepted.
>
> ---
>
> 2. *How does the LD method benefit when there is existing domain knowledge about an environment?*
>
> Great question! Domain knowledge can help a system designer to provide useful (ideally Markovian) features to the decision-making agent. If the system designer wants to check whether these features are in fact Markovian, he or she could use the LD to test for it.
>
> Interestingly, even when the environment is fully observable, the λ-discrepancy may still be useful, because function approximation can sometimes act like partial observability. If the environment has Markovian observations but these observations are sent through a lossy function approximator (like a neural network), then minimizing λ-discrepancy could help the function approximator to learn better features.
>
> ---
>
> 3. *How is the LD method affected when switching between online and offline-RL settings?*
>
> Another good question! For offline RL, LD needs to be applied with care when the data is not generated from the policy being evaluated, TD(λ) no longer has the same fixed point for all values of λ under different policies. Our theory relating the Markov assumption to LD would not hold between two value functions under different policies. However, one could use tools from off-policy RL such as importance sampling to adapt our methods to this new setting. Another idea is to minimize the LD for the data-generating policy instead of the evaluation policy. Due to the result in Theorem 1, this should result in a memory function that, in theory, mitigates partial observability for other policies as well. This is similar to what we do in the analytical experiments in Sec. 4. That setup is akin to an offline policy evaluation approach for memory learning, which is an exciting avenue for future work.

---

> > ### Comment · Reviewer_pbjp · 2024-08-13
> >
> > A thanks to the authors for providing a thorough rebuttal, which lead me to increase my score.
> > - The global rebuttal as well as the direct one were detailed and addressed many points that were raised.
> > - As suggested, the authors performed an experiment comparing a gradual shift from a MDP to POMDP setting and will include it in their final submission.
> > - Further, the authors will provide information about the choosing of the $\lambda_1$ and $\lambda_2$ hyperparameters.
> > - Also, the authors arguments on environment complexity were convincing; to demonstrate that the LD is helpful is done by the experiments provided. However, to demonstrate that this is true in practice does require experiments with more complex scenarios.
> > - Including the architecture diagram is also very valuable and will help readers to understand the training procedure better. (I would recommend using small labels such as 1, 2 and 3 in the right part of the figure which could be referenced in the caption.)

---

### Author Rebuttal · Authors · 2024-08-07

Thank you all for your feedback and suggestions. Your reviews helped us improve the quality of our submission.

We've made the following requested changes (**see PDF**):
1. Experiment varying “Markovness” of environment (new Fig 1)
2. Visual comparison of learned memories for PacMan (new Fig 2, right)
3. New results for Parity Check including λ-discrepancy (LD) detection (new Fig 3, left) and memory optimization (new Fig 3, right)
4. Architecture diagram to clarify training procedure and RNN results (new Fig 2, left)
5. Simplified Sec. 3, including discussion of edge cases (see response to reviewer **8yd3**)

We also justify our choice of environments and discuss selecting λ values below.

---

**Varying "Markovness" (pbjp, 8yd3)**

We ran an additional experiment to measure LD while interpolating between an MDP and a POMDP for the T-Maze environment. We interpolate between an identity observation function (in which each state leads to a distinct observation) and an aliased one (where certain states cannot be distinguished). In our experiment, the LD has a minimum at zero for MDP and increases with partial observability for three different types of state aliasing. The results are in new Fig 1, and we will add them to the camera-ready if the paper is accepted.

Note: this experiment necessarily compares LD between different environments, which it was not designed for. This can lead to counterintuitive results depending on discount factor and choice of weighted norm, since the size of the observation space is changing, and states from later in the trajectory become aliased with earlier ones. To mitigate these issues, (for this experiment only) we use the max norm and set γ≈1.

---

**Memory visualization**

We also added visualizations of the agent's memory functions. In PacMan, the food dots provide a convenient way to track the agent's path. By training a small neural network probe (offline) to predict the remaining food dot locations from the agent’s memory, we can visualize how well the agent remembers where it has been. We compare the memory of a baseline RNN agent against that of an RNN with LD minimization, and find that minimizing LD facilitates much more accurate reconstruction of the agent’s path. These visualizations are shown in new Fig. 2 (right).

---

**Parity Check (rpiT)**

While our analysis showed that symmetric environments like the Parity Check example are likely rare, our initial submission did not offer a practical mitigation strategy for them. However, it turns out such symmetries are actually easy to handle: we can simply add a small amount of (stochastic) memory. If the environment is truly an MDP, adding memory will have no effect on LD, but if it is a POMDP, a randomly initialized memory function may be enough to break symmetry and produce a LD>0. Indeed, almost all of the 10K random memory functions we generated led to substantial LD (Fig 3, left). A subsequent memory optimization (using the procedure in Sec. 4) was able to extract 85% of the normalized return from the environment (Fig 3, right). This helps explain why such edge cases are not an issue for our RNN experiments: random memory functions (like the ones we initialize with in Sec. 5) break the symmetries that would appear in POMDPs with LD=0.

---

**Environment Complexity (pbjp, 8yd3)**

Some reviewers argued that the environments we tested were “simple” or “low complexity”. Our objective was to test complexity in the required memory function. For this reason, we intentionally avoided high-dimensional observation spaces that would require substantial representation learning on top of the memory learning task we were investigating. While the observation spaces of the domains in Sec. 5 may seem “simple”, the memory functions required to solve these domains are very complex, as reviewer 8yd3 pointed out, “I appreciate the experiments on larger POMDP planning benchmarks, since they require complex memory that seem non-trivial to learn.” For example, in P.O. PacMan, the agent must remember its current position, track which food dots and power-ups it has consumed, and remember in which directions it has recently seen ghosts. This is a highly challenging memory task and is sufficient to make our scientific point, since baseline RNNs struggle to learn a useful memory function (see new Fig. 2, right). We agree that investigating larger domains is important for future work, but we do not want to obscure the point we make here by conflating memory learning with arbitrary feature learning.

---

**Understanding and selecting λ parameters (pbjp, AeCz, 8yd3)**

Recall that TD(λ) is a weighted average of n-step returns, where λ controls the weighting scheme. Including a bootstrapped value term in the return target ($… + γ^n V(ω_{t+n})$) implicitly treats the environment like an MDP, as if the observation $ω_{t+n}$ is a Markovian state. Low values of λ give more weight to shorter rollouts where the bootstrapped term is discounted less (and thus has greater significance), whereas larger λ values put more weight on longer rollouts where the bootstrapped term is more heavily discounted. With two distinct λs, we can see if the estimators diverge when they use bootstrapping components of differing strength (which does not happen for MDPs).

Our theoretical and empirical results indicate that $λ_1$ and $λ_2$ should be well separated to provide sufficient signal for learning memory. Indeed, our hyperparameter sweep in Sec. 5 selected λ parameters that reflect this intuition, with a large difference between $λ_1$ and $λ_2$ values. In principle, LD is maximized for λ in {0,1}, but in practice, setting λ too close to 0 can lead to high bias, especially during the early part of learning, and setting λ too close to 1 can lead to high variance in returns and instability in the estimator. This is why our experiments only test λ values that are in the range [0.1, 0.95].

We will add this discussion to the paper if it is accepted.

---

### Author Response · Authors · 2024-08-14

Thank you all for the productive discussions! We appreciated the collaborative spirit of your reviews, and your feedback and suggestions were very helpful. We are happy to hear that we were able to address your main concerns and that you all now recommend acceptance. We will incorporate your latest feedback into the camera ready if the paper is accepted.

You identified several interesting follow-up directions that we are excited to pursue, and we are glad that you see the potential for this work to have such broad impact.

Sincerely,

Submission12090 authors

---

### Decision · Program_Chairs · 2024-09-25

**Decision:**

Accept (poster)

**Comment:**

This paper tackles reinforcement learning problems under partial observability. The paper proposes a new metric to build state representation based on the TD($\lambda$), which makes the novelty of the method is clear. While the empirical analysis still seems limited, missing a comparison with state-of-the-art RL methods for POMDPs, the paper provides sufficient theoretical support for the proposed approach. Considering all reviewers are positive about the paper, I recommend its acceptance.